



# A framework for deriving drought indicators from GRACE

Helena Gerdener[1], Olga Engels[1], and Jürgen Kusche[1]

[1]Institute of Geodesy and Geoinformation, University of Bonn, Bonn, Germany.

**Correspondence:** Helena Gerdener (gerdener@geod.uni-bonn.de)

**Abstract.** Identifying and quantifying drought in retrospective is a necessity for better understanding drought conditions and the propagation of drought through the hydrological cycle, and eventually for developing forecast systems. Hydrological droughts refer to water deficits in surface and subsurface storage, and since these are difficult to monitor at larger scales, several studies have suggested to exploit total water storage data from the GRACE (Gravity Recovery and Climate Experiment) satellite gravity mission to analyse them. This has led to the development of GRACE-based drought indicators. However, it is unclear how the ubiquitous presence of climate-related or anthropogenic water storage trends, which has been found from GRACE analyses, masks drought signals. Thus, this study aims at a better understanding of how drought signals, in the presence of trends and GRACE-specific spatial noise, propagate through GRACE drought indicators. Synthetic data are constructed and existing indicators are modified to possibly improve drought detection. Our results indicate that while the choice of the indicator should be application dependent, larger differences in robustness can be observed. We found a modified, temporally accumulated version of the Zhao et al. (2017) indicator in particular robust under realistic simulations. We show that trends and accelerations seen in GRACE data tend to mask drought signals in indicators, and that different spatial averaging methods required to suppress the spatially correlated GRACE noise affect the outcome. Finally, we identify and analyse two droughts in South Africa using real GRACE data and the modified indicators.

## 1 Introduction

Droughts are recurrent natural hazards that affect environment and economy with potentially catastrophic consequences. Drought impacts reach from reduced streamflow, water scarcity, and reduced water quality to increased wildfires, soil erosion and increased quantities of dust, crop failure and large-scale famine. With climate change and population growth, frequency

20 and impact of droughts are projected to increase for many regions of the world (IPCC, 2013). Drought types can be distinguished depending on their effect on the hydrological cycle (e.g. Changnon, 1987; Mishra and Singh, 2010). In this study we focus on hydrological drought, a multiscale problem which may last weeks or many years, and which may affect local or continental regions. For example, the severe drought between mid-2011 and -2012 affected millions in the entire East Africa region (Somalia, Djibouti, Ethiopia and Kenya) and let to famine with an estimate of 258,000 deaths (Checchhi and Robinson,





2013). From 2012 to 2016, the US state of California experienced a historical drought that adversely affected groundwater levels, forests, crops, fish populations, and led to widespread land subsidence (Mann and Gleick, 2015; Moore et al., 2016). In contrast, European droughts as e.g. in 2018 typically last a few months in exceptionally dry summers. For South Africa, due to a complex rainfall regime, areas and percentage of land surface affected by drought can vary strongly (Rouault and Richard,
2005) and their identification depends on the accumulation period considered.

Hydrological drought refers to a deficit of accessible water, i.e. water in natural and man-made surface reservoirs and subsurface storages, with respect to normal conditions. The propagation of drought through the hydrological cycle typically begins with a lack of precipitation, developing to runoff and soil moisture deficit, followed by decreasing streamflow and groundwater levels (Changnon, 1987). However, no unique standard procedures exist for measuring the deficit and for defining
the normal conditions. In order to arrive at operational definitions, e.g. for triggering a response according to drought class, a large variety of drought indicators has been defined which typically seek to extract certain sub-signals from observable fields (Bachmair et al., 2016; Wilhite, 2016; Mishra and Singh, 2010; Van Loon, 2015). Reviews of hydrological drought indicators are contained in Keyantash and Dracup (2002); Wilhite (2016); Mishra and Singh (2010); Tsakiris (2017). Streamflow is the most frequently used observable in these studies.

Drought detection is mostly restricted to single fluxes (precipitation or streamflow) or storages (surface soil moisture, reservoir levels) that are easy to measure. Much less measurements are available to assess water content in deeper soil layers and groundwater storage deficit, or the total of all storages. The NASA/DLR Gravity Recovery and Climate Experiment (GRACE) satellite mission, launched in 2002, has changed this situation since GRACE-derived monthly gravity field models can be converted to total water storage changes (TWSC, Wahr et al., 1998). GRACE consisted of two spacecraft following each other and
linked with an ultra-precise microwave ranging instrument; these ranges are routinely processed to monthly gravity models and further to mass change maps. Since other mass transports in atmosphere and ocean are removed during the processing, GRACE indeed provides quantitative measure of surface and subsurface water storages (Chen et al., 2009; Frappart et al., 2013). Meanwhile, GRACE has been continued with the GRACE-FO mission and first data is expected to become available in May 2019.

Studies of drought detection with GRACE TWSC can be summarized in three groups: (i) using monthly maps of TWSC directly, (ii) partitioning TWSC timeseries into sub-signals that include drought signatures, or (iii) using indicators. For example, Seitz et al. (2008) investigated the 2003 heat wave over seven Central European basins using GRACE timeseries; they found good agreement to net precipitation minus evaporation. Other studies focused on drought detection using TWSC sub-signals, e.g. trends were used to identify drought in Central Europe (Andersen et al., 2005) and for the Tigris-Euphrates-Western Iran
(Voss et al., 2013). After decomposing GRACE TWSC into a seasonal and non-seasonal signal, Chen et al. (2009) were able to detect the 2005 drought in the Central Amazon river basin while Zhang et al. (2015) identified two droughts in 2006 and 2011 in the Yangtze river basin. In the latter study, the El Niño/Southern Oscillation (ENSO) was identified as a possible driver for drought events in the Yangtze river basin. However, neither GRACE nor GRACE-FO enable one to separate different compartments such as groundwater storage without utilizing external information, and their spatial (about 300 km for GRACE) and
temporal (nominally one month) resolution are limited. Several efforts are therefore focusing on assimilating GRACE TWSC





maps into hydrological or land surface models (e.g., Zaitchik et al., 2008; Eicker et al., 2014; Girotto et al., 2016; Springer, 2019).

Thus perhaps not surprisingly, a number of GRACE-based drought indicators have been suggested (e.g. Houborg et al., 2012; Thomas et al., 2014; Zhao et al., 2017), typically either based on e.g. normalization or percentile rank methods. However, a

comprehensive comparison and assessment of these indicators is still missing, in particular in the presence of (1) trend signals as picked up by GRACE in many regions that may reflect non-stationary 'normal' conditions, (2) correlated spatial noise that is related to GRACE, and (3) the inevitable spatial averaging applied to GRACE results to smooth out noise (Wahr et al., 1998). From a water balance perspective, GRACE TWSC variability mainly represents monthly total precipitation anomalies (e.g., Chen et al., 2010; Frappart et al., 2013). It is thus obvious that GRACE drought indicators will contain signatures that

are visible in meteorological drought indicators, yet the difference should tell about the magnitude of other contributions (e.g. increased evapotranspiration due to radiation) to hydrological drought.

Fig. 1 shows a time series of region-averaged, de-trended and de-seasoned GRACE water storage changes over Eastern Brazil (Ceará state) compared to the region-averaged 6 months Standard Precipitation Indicator SPI (McKee et al., 1993) to illustrate the potential of GRACE TWSC for drought monitoring. As can be expected, TWSC and 6 months SPI appear mod-

erately similar (correlation 0.43), characterised by positive peaks e.g. at the beginning of 2004 and at the end of 2009, and negative peaks at the beginning of 2013. This motivates us to modify common GRACE indicators to account for accumulation and differencing periods. To our knowledge, this is the first study where (modified) indicators are tested in a synthetic framework based on a realistic signal that includes a hypothetical drought. We hypothesize that in this way we can (i) assess indicator robustness, with respect to identifying a 'true' drought of given duration and magnitude, and (ii) understand how trend

signals and spatial noise propagate into indicators and mask drought detection. In addition, we investigate to what extent the spatial averaging that is required for analysing GRACE data affects indicators. For this, we compare spatially average gridded indicators to indicators derived from spatial averaged TWSC.

This contribution is organized as follows: in section 2 we will review three GRACE-based drought indicators and modify them to accommodate either multi-month accumulation or differencing, while in section 3 our framework for testing GRACE

indicators in a realistic simulation environment will be explained. Then, section 4 will provide simulation results and finally the results from real GRACE data. A discussion and conclusion will close the paper.

## 2   Indicators for hydrological drought

Hydrological drought indicators are mostly based on observations of single water storages or fluxes, e.g. for precipitation, snowpack, streamflow, or groundwater. In general, indicator definitions can be arranged in four categories: 1) data normaliza-

tion, 2) threshold-based, 3) quantile scores and 4) probability-based (e.g., Zargar et al., 2011; Keyantash and Dracup, 2002; Tsakiris, 2017).

Since total water storage deficit may be viewed as a more comprehensive information for drought, with the advent of GRACE total water storage changes (TWSC) data new indicators have been developed. For example, Frappart et al. (2013)



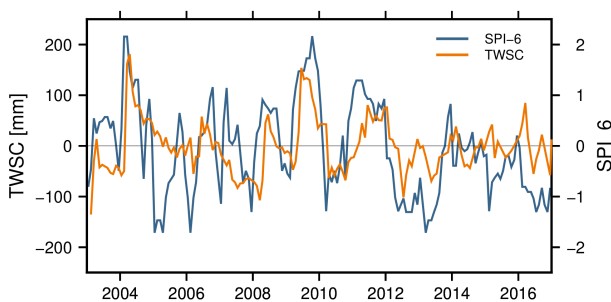

**Figure 1.** De-trended and de-seasoned GRACE TWSC [mm](orange) and the SPI[-] of 6-month accumulated precipitation (blue), spatially averaged for Ceará, Brazil.

developed a drought indicator based on yearly minima of water storage and a standardization method, and Kusche et al. (2016) computed recurrence times of yearly minima through generalized extreme value theory. Other indicators explored the monthly resolution of GRACE, e.g. the Total Storage Deficit Index (TSDI, Agboma et al., 2009), the GRACE-based Hydrological Drought index (GHDI, Yi and Wen, 2016), the Drought Severity Index (DSI, Zhao et al., 2017), and the Drought

Index (DI, Houborg et al., 2012). Further, Thomas et al. (2014) presented a water storage deficit approach to detect drought magnitude, duration, and severity based on GRACE-derived TWSC. To our knowledge, only the Zhao et al. (2017), Houborg et al. (2012), and Thomas et al. (2014) methods are able to detect drought events from monthly GRACE data without any additional information. Therefore, these three indicators will be discussed further.

In order to stress the link between GRACE-based and meteorological indicators, we first describe the relation of TWSC and

precipitation. Assuming evapotranspiration ($E$) and runoff ($Q$) vary more regular as compared to precipitation (i.e. $\Delta E = 0$, $\Delta Q = 0$), the monthly GRACE TWSC ($\overline{\Delta s}$) corresponds to precipitation anomalies ($\Delta P$) accumulated since the GRACE storage monitoring began

$$\overline{\Delta s}(t) = \Delta t \sum_{t_0}^{t} \Delta P \ , \tag{1}$$

where $\Delta t$ is the time from $t_0$ to $t_1$. In contrast, the difference between GRACE months

$$\overline{\Delta s}(t_2) - \overline{\Delta s}(t_1) = \Delta t \sum_{t_1}^{t_2} \Delta P \tag{2}$$

corresponds to the precipitation anomaly accumulated between these months. Accumulating monthly TWSC corresponds thus to an iterative summation over the precipitation anomalies described by

$$\sum_{t_0}^{t} \overline{\Delta s}(t) = \Delta t \sum_{\tau = t_0}^{t} \sum_{t_0}^{\tau} \Delta P. \tag{3}$$

In the following, we will discuss and extend the definition of Zhao et al. (2017), Houborg et al. (2012), and Thomas et al.

(2014) GRACE-based indicators, which are then referred to as the Zhao-method, Houborg-method, and Thomas-method.



## 2.1 Zhao-method

In the approach of Zhao et al. (2017), one considers GRACE-derived monthly gridded TWSC for $n$ years,

$$x_{i,j} = \overline{\Delta s}(t_{i,j}) \tag{4}$$

with

$$t_{i,j} = i + \left(j - \frac{1}{2}\right)\frac{1}{12} \qquad i = 1,\ldots,n \qquad j = 1,\ldots,12 . \tag{5}$$

Let us define the monthly climatology, i.e. mean monthly TWSC, $\tilde{x}_j$ with $j = 1,\ldots,12$ and the standard deviation $\tilde{\sigma}_j$ of the anomalies in month $j$ with respect to the climatological value as

$$\tilde{x}_j = \frac{1}{n}\sum_{i=1}^{n} x_{i,j} \tag{6}$$

$$\tilde{\sigma}_j = \left(\frac{1}{n}\sum_{i=1}^{n}(x_{i,j} - \tilde{x}_j)^2\right)^{1/2} . \tag{7}$$

Zhao et al. (2017) define their drought severity index 'GRACE-DSI' as the standardized anomaly

$$\text{TWSC-DSI}_{i,j} = \frac{x_{i,j} - \tilde{x}_j}{\tilde{\sigma}_j} \tag{8}$$

of a given month $t_{i,j}$ and provide a scale from -2.0 (exceptional drought) to +2.0 (exceptional wet), as shown in Tab. 1. There is no particular probability distribution function (PDF) underlying the method, however if we assume the anomalies for a given month follow a Gaussian PDF it is straightforward to compute the likelihood of a given month falling in one of the Zhao et al. (2017) severity classes: For example, 2.1 % of months would be expected to turn out as exceptional drought and 2.1 % as exceptionally wet. This can be applied to any other PDF.

   Drought severity, however, should be related to the duration of a drought. For example McKee et al. (1993) showed how typical time scales of 3, 6, 12, 24, and 48 months of precipitation deficits are related to their impact on usable water sources. To account for the relation between severity and duration in the Zhao et al. (2017) approach, we consider $q$-months accumulated TWSC, which is approximately related to precipitation in Eq. (3) as

$$x_{i,j,q}^{+} = \sum_{k=1}^{q} \overline{\Delta s}(t_{i,j+1-q}) \tag{9}$$

with $t_{i,j+1-q} = t_{i-1,j+13-q}$ for $j + 1 - q < 1$ or equivalently written for $q$-months averaged TWSC

$$x_{i,j,q}^{+} = \frac{1}{q}\sum_{k=1}^{q} \overline{\Delta s}(t_{i,j+1-q}) . \tag{10}$$

For example for $q = 3$, we would look for the 3 months running mean Dec-Jan-Feb, Jan-Feb-Mar, and so on. In the next step, one computes e.g. the climatology and anomalies as with the original method. On the other hand, we can relate hydrological to





meteorological indicators using Eq. (2). To develop a TWSC indicator that can be compared to indicators based on accumulated precipitation, one should rather consider the $q$ months differenced TWSC

$$x_{i,j,q}^- = \overline{\Delta s}(t_{i,j}) - \overline{\Delta s}(t_{i,j+1-q}).$$ (11)

Thus, equivalent to TWSC-DSI$_{i,j}$ in Eq. (8), through standardization we can define two new multi-month indicators (TWSC-
DSIA and TWSC-DSID) by using accumulated (A) and differenced (D) TWSC (Eq. 9 and 11) as

$$\text{TWSC-DSIA}_{i,j,q} = \frac{x_{i,j,q}^+ - \tilde{x}_{j,q}^+}{\tilde{\sigma}_{j,q}^+}$$ (12)

and

$$\text{TWSA-DSID}_{i,j,q} = \frac{x_{i,j,q}^- - \tilde{x}_{j,q}^-}{\tilde{\sigma}_{j,q}^-}.$$ (13)

Finally, it is obvious that sampling the full climatological range of dry and wet months is not yet possible with the limited
GRACE data period. Therefore, Zhao et al. (2017) suggest applying a bias correction to avoid the under- or overestimation of drought events. This implies using TWSC from multi-decadal model runs, which is feasible but not in the focus of this study.

**Table 1.** Drought severity level of the TWSC-DSI (Zhao et al., 2017). The values of TWSC-DSI are unitless.

| | TWSC-DSI [-] | |
| --- | --- | --- |
| **Drought Severity Level** | **Min.** | **Max.** |
| Abnormal | $-0.8$ | $-0.5$ |
| Moderate | $-1.3$ | $-0.8$ |
| Severe | $-1.6$ | $-1.3$ |
| Extreme | $-2.0$ | $-1.6$ |
| Exceptional | | $-2.0$ |

## 2.2   Houborg-method

Houborg et al. (2012) define the drought indicator 'GRACE-DI' via the percentile of a given month, $t_{i,j}$, with respect to the cumulative distribution function (CDF). The GRACE-DI is applied to TWSC by

$$\text{TWSC-DI}_{i,j} = \frac{\sum_i (x_j \le x_{i,j})}{\sum_i x_j} \cdot 100,$$ (14)

i.e. all years containing month $j$ are counted for which TWSC is equal or lower than TWSC in month $j$ and year $i$, and normalized by the number of the years that contain month $j$. The indicator value is assigned to five severity classes as shown in Tab. 2. For example, exceptional droughts occur up to 2 % of the entire time period at any location.





Again, to relate drought severity to duration, we proceed to multi-month accumulation (Eq.9) and differences (Eq.11) resulting in the definition of two new indicators based on TWSC-DI$_{i,j}$ in Eq. (14):

$$\text{TWSC-DIA}_{i,j} = \frac{\sum_i (x^+_{j,q} \leq x^+_{i,j,q})}{\sum_i x^+_{j,q}} \cdot 100 \qquad (15)$$

$$\text{TWSC-DID}_{i,j} = \frac{\sum_i (x^-_{j,q} \leq x^-_{i,j,q})}{\sum_i x^-_{j,q}} \cdot 100. \qquad (16)$$

Assuming again the CDF equals to the cumulative Gaussian, for example 0.6 % of months would be detected as exceptionally dry or 9.5 % of months as abnormally dry. Houborg et al. (2012) applied the percentile approach also separately to surface soil moisture, root zone soil moisture and groundwater storage, which were derived by assimilating GRACE-derived TWSC into a hydrological model, and the CDFs were adjusted to a long-term model run. Here, we focus on TWSC from GRACE only and, as explained in Sec. 2.1, we therefore disregard the bias correction.

**Table 2.** Drought severity level of the TWSC-DI (Houborg et al., 2012). The values of TWSC-DI are given in %.

| | TWSC-DI [%] | |
|---|---|---|
| **Drought Severity Level** | **Min.** | **Max.** |
| Abnormal | 20 | 30 |
| Moderate | 10 | 20 |
| Severe | 5 | 10 |
| Extreme | 2 | 5 |
| Exceptional | 0 | 2 |

### 2.3 Thomas-method

Thomas et al. (2014) define a drought by considering the number of consecutive months below a threshold. Given TWSC observations $x_{i,j}$ and a threshold $c$, we can compute anomalies by

$$\Delta x_{i,j} = \begin{cases} 0 & \text{for } x_{i,j} \geq c \\ x_{i,j} - x_j & \text{for } x_{i,j} > c. \end{cases} \qquad (17)$$

The threshold can be derived following different concepts, however, Thomas et al. (2014) use the monthly climatology $x_j$ (Eq. 6). Here, we also consider using a fitted signal for defining the threshold. The signal is computed by

$$x(t) = a_0 + a_1(t - t_0) + a_2 \frac{1}{2}(t - t_0)^2 + b_1 \cos(\omega t) + b_2 \sin(\omega t) + c_1 \cos(2\omega t) + c_2 \sin(2\omega t) \qquad (18)$$

at time $t$ with a constant $a_0$, linear trend $a_1$ and acceleration $a_2$ terms, an annual signal $b_1$ and $b_2$, and similar for a semi-annual signal $c_1$ and $c_2$. The Thomas-method then identifies drought events through the computation of magnitude, duration,





and severity: the magnitude or water storage deficit equals to $\Delta x_{i,j}$ (Eq. 17) and the duration $d_{i,j}$ is given by the number of consecutive months where TWSC is below the threshold. Thomas et al. (2014) propose a minimum number of 3 consecutive months that are required for the computation of drought duration. By using the deficit $\Delta x_{i,j}$ and the duration $d_{i,j}$, the severity $s_{i,j}$ of the drought event can finally be computed by

$$s_{i,j} = \Delta x_{i,j} d_{i,j} . \tag{19}$$

Severity is therefore a measure of the combined impact of the water storage deficits and duration, see Thomas et al. (2014) and Humphrey et al. (2016).

## 3   Framework to derive synthetic TWSC for computing drought indicators

### 3.1   Methods

In order to analyse the performance of drought indicators, we suggest to construct a synthetic timeseries of 'true' total water storage changes (TWSC), on a grid, first. We base our drought simulations on the GRACE data model

$$\Delta s(t) = x(t) + \eta(t) + \epsilon(t) \tag{20}$$

including the in Sec. 2.3 introduced signal $x$ (Eq. 18) (constant, linear and time varying trend, and seasonality), an interannual signal $\eta$, which will carry the simulated 'true' drought signature and which has been de-trended and de-seasoned, and a GRACE-specific noise term $\epsilon$. To simulate the 'true' signal as realistically as possible using Eq. (20), we first analyse real GRACE-TWSC following the steps summarized in Fig. 2. We derive 1) the signal components constant, trend, acceleration, annual, and semi-annual sine wave, 2) temporal correlations, 3) a representative drought signal quantified by strength and duration, and 4) spatially correlated noise, the latter from GRACE error covariance matrices. While the first three steps are generic and can be used for simulating other observables, step 4 is directly related to the measurement noise, in this case the GRACE noise.

As an input to the simulation, GRACE-TWSC are derived by mapping monthly ITSG-GRACE2016 gravity field solutions of degree and order 60, provided by TU GRAZ (Mayer-Gürr et al., 2016), to TWSC grids. As per standard practice, we add degree-one spherical harmonic coefficients from (Swenson et al., 2008) and degree 2, order 0 coefficients from laser ranging solutions, (Cheng et al., 2011). Then, we remove the temporal mean field, apply a DDK3-filtering (Kusche et al., 2009) to suppress excessive noise, and map coefficients to TWSC via spherical harmonic synthesis. We also remove the effect of ongoing glacial isostatic adjustment (GIA) following A et al. (2013).

Droughts are a multiscale phenomenon, and for a realistic simulation we must first define the largest spatial scale to which we will apply the model of Eq. (20). In other words, we first need to identify coherent regions in the input data for which our approach is then applied at grid-scale prior to step 1. For this, we apply two consecutive steps: we first compute temporal signal correlations by fitting an autoregressive (AR) model (Appendix A; Akaike, 1969) to detrended and deseasoned GRACE data. These TWSC residuals contain interannual and subseasonal signals including real drought information. Temporal correlation





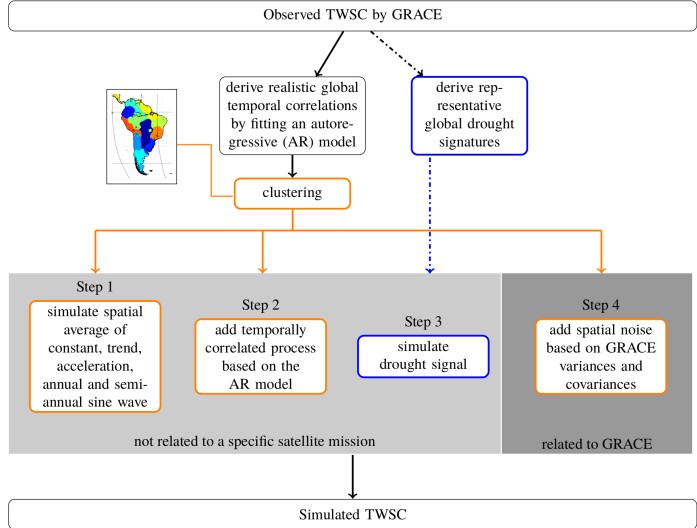

**Figure 2.** Concept of synthetic framework to generate synthetic TWSC

coefficients are then used as input for an Expectation Maximization (EM) clustering (Dempster et al. (1977), Redner and Walker (1984)), because regions with similar residual TWSC correlation within the interannual and subseasonal signal are hypothesized here to be more likely affected by the same hydrological processes. The EM algorithm by Chen (2018) is modified to identify regional clusters by maximizing the likelihood of the data (Alpaydin, 2009).

As a result of this procedure, we chose three clusters located in East Brazil (EB), South Africa (SA), and West India (WI), which were also affected by droughts in the past (e.g. Parthasarathy et al., 1987; Rouault and Richard, 2003; Coelho et al., 2016). The following simulation steps are then applied to each of these three clusters.

In step 1 we estimate the signal coefficients according to Eq. (18) through least squares fit for each grid cell within the cluster. The coefficients are then spatially averaged to create a signal representative for the mean conditions within the region, and they

are used to create the constant, trends, and the seasonal part of synthetic time series. To simulate realistic temporal correlations at the region scale (step 2), we use the AR-model identified beforehand (Fig.2) and again average AR-model coefficients within the cluster. Then, we apply an AR model with the estimated optimal order and the averaged correlation coefficient (Eq. A1) to the synthetic time series to add temporal correlations.

Simulating realistic drought events in step 3 is challenging because, to our knowledge, no unique procedure to simulate

realistic drought periods for TWSC exists. For this reason, we first perform a literature review to identify representative drought periods and magnitudes for selected regions. Among others, this includes the 2003 European drought and the drought in the Amazon basin in 2011 (e.g., Seitz et al., 2008; Espinoza et al., 2011). TWSC within the identified drought period are then eliminated from the time series. In the next step, the parameters describing the constant, trend, acceleration and seasonal signal before and after the drought are used to 'extrapolate' these signals during the drought period. By computing the difference of





the original GRACE-TWSC time series and the continued signal in the drought period, we can separate non-seasonal variations from the data, which represent the drought magnitude. Our hypothesis is that the non-seasonal variations that we derive from the procedure possibly show a systematic behaviour that can be parameterized. To extract this systematic behaviour, all extracted droughts are transformed to a standard duration. To compare the different drought signals, a standard duration and a standard magnitude are arbitrarily set to 10 months and -100 mm, respectively. Finally, a synthetic drought signal $\eta$ is generated by using the extracted knowledge of drought duration, drought magnitude and systematic behaviour and it is added to the synthetically generated signal (Eq. 20).

In step 4 we add GRACE-specific spatially correlated and temporally varying noise $\epsilon$ (Eq. 20). First, for each month $t$ we extract a full variance-covariance matrix $\Sigma$ for the region grid cells from GRACE-TWSC. Next, whenever $\Sigma$ is positive definite, we apply Cholesky decomposition $\Sigma = R^T R$, while if $\Sigma$ is only positive semi-definite we apply eigenvalue decomposition (Appendix B). Second, we generate a Gaussian noise series $v$ of the length $n$, where $n$ represents the number of grid cells within the cluster. Finally, spatial noise in month $t$ is simulated through

$$\epsilon = R^T v. \tag{21}$$

The final synthetic signals for each grid cell within a cluster will thus exhibit the same constant, trend, acceleration, seasonal signal, temporal correlations, and drought signal, but spatially different and correlated noise. In the following, we will test the hypothesis that GRACE indicators depend on the presence of trend and random input signals using the generated synthetic time series.

We believe that our synthetic framework based on real GRACE data has multiple benefits: i) we are able to identify the skill of an indicator by comparing the 'true' drought duration and magnitude (step 3) to the indicator results; ii) we are able to detect the influence of other typical GRACE signals on the drought detection; iii) comparing different indicator outputs allows us to identify the most suitable indicator for a specific application.

## 3.2 Synthetic TWSC

Here, we will briefly discuss the TWSC simulation following methods described in the previous section.

When estimating AR models for detrended and deseasoned global GRACE data, we find that for more than 70 % of the global land TWSC grids are best represented by an AR(1) process (App. Fig. A1). Therefore, we apply the AR(1) model for each grid. Fig. 3 shows the estimated AR-model coefficients, which represent the temporal correlations, ranging from very low up to 0.3, e.g. over the Sahara or in South West Australia, to about 0.8, for example in Brazil or in South Eastern U.S. EM-clustering is then based on these coefficients.

The selected three clusters (Fig. 3) show differences between the signal coefficients of the functional model (step 1, Eq. 18), which are exemplarily shown for the linear trend. We find a mean linear trend for the East Brazil cluster of 1.0 mm TWSC per year; South Africa shows a higher trend of 5.0 mm per year and for West India the trend is 56.3 mm per year (Tab. 3). The trends for East Brazil and South Africa in GRACE TWCS have been identified before (e.g. Humphrey et al., 2016; Rodell et al., 2018). We did not find confirmations for the strong linear trend in West India, e.g. Humphrey et al. (2016) identified about 7





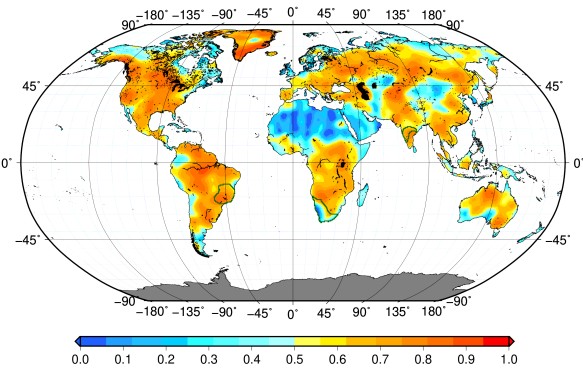

**Figure 3.** AR(1)-model coefficients [-] for global GRACE-TWSC. The polygons of the clusters of East Brazil, South Africa and West India are added in dark green.

mm per year within this region. We assume that in this study the linear trend for West India is estimated as strong positive because we additionally identify a strong negative acceleration of -8.03 mm per year$^2$ in West India. However, our simulation will cover weak and strong trends. In fact, all coefficients show such strong differences, which suggests that we cover different hydrological conditions when simulating TWSC for the three regions. In step 2 we identify correlations of 0.74 in East Brazil,

**Table 3.** Coefficients for signals contained in GRACE-TWSC that were extracted within the clusters of East Brazil, South Africa, and West India. These coefficients are used to simulate synthetic TWSC.

| Cluster | Constant | Linear Trend | Acceleration | Annual | | Semi-annual | | AR-correlation |
|---|---|---|---|---|---|---|---|---|
| East Brazil | 34.85 | 1.02 | -1.77 | 6.83 | 106.12 | 4.69 | 9.47 | 0.74 |
| South Africa | -24.00 | 4.98 | -0.38 | -4.31 | -2.34 | -1.23 | 1.07 | 0.42 |
| West India | -139.37 | 56.30 | -8.03 | 30.23 | -122.69 | -24.22 | 25.24 | 0.79 |

5  0.79 in West India, and 0.42 in South Africa (Tab. 3).

Searching for drought duration and magnitude (step 3) led to four droughts seen in GRACE-TWSC: The 2005 and 2010 droughts in the Amazon (e.g. Chen et al., 2009; Espinoza et al., 2011), the 2011 drought in Texas (e.g. Long et al. (2013)), and the 2003 drought in Europe (e.g. Seitz et al. (2008)). To extract the drought duration, we compared drought begin and end in these and other papers. We found that different studies do not exactly match, with inconsistencies likely due to different

10  methodologies used. Furthermore, some authors only specified the year of drought. Droughts finally extracted from the literature had a duration of 3 to 10 months (Fig. 4a-d). Unless otherwise specified, we decided to base our simulations on a duration of 9 months to represent a clear identifiable drought duration. Extracted drought magnitudes range from about -20 to -350 mm TWSC (Fig. 4a-d). Therefore, in order to simulate a drought magnitude that has a clear influence on the synthetic time series, we set the magnitude to -100 mm.



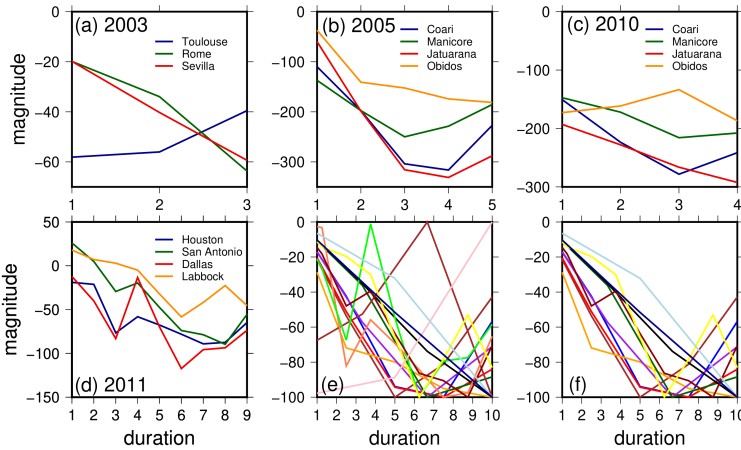

**Figure 4.** Extracted drought periods from GRACE-TWSC for the droughts in (a) Europe 2003, (b) Amazon river basin 2005, (c) Amazon river basin 2010, (d) Texas 2011. (e) All droughts from (a-d) were transformed to standard severity and duration. (f) as (e) but after removing four timeseries with a significant different temporal behaviour.

As described in Sec. 3.1, we transform these water storage droughts to a standard duration and magnitude to understand whether a typical signature can be seen. However, Fig. 4e remains inconclusive as in particular four standardized droughts show a very different temporal behaviour: Toulouse in 2003, Obidos in 2010, and Houston and Dallas in 2011. When we remove those four timeseries (Fig. 4f), a systematic behaviour can be identified and parameterized using a linear or quadratic

temporal model. However, seen these difficulties, we decided to stick to the most simple TWSC drought model, i.e. a constant water storage deficit within a given time span.

In step 4, we project the simulation on a $0.5°$ grid and add spatially correlated GRACE noise. A few representative time series of the gridded synthetic total water storage change are shown in Fig. 5 for East Brazil (EB), South Africa (SA), and West India (WI) for the GRACE time period from January 2003 to December 2016. The effect of realistic GRACE noise (dark

blue vs. light blue) is clearly visible, in particular for the SA case with low annual amplitude. The synthetic drought period is placed from January to September 2005 (light brown) in all three regions. Synthetic TWSC variability includes considerable (semi-) annual variations for EB based on Tab. 3. Furthermore, a strong trend and acceleration is contained in the synthesized time series for East Brazil and West India (Tab. 3).

## 4  Indicator-based drought identification with synthetic and real GRACE data

### 4.1  Synthetic TWSC: masking effect of trend and seasonality

Here, we analyse how non-drought signals, such as a linear or accelerated water storage trend and the ubiquitous seasonal signal, propagate through the Zhao-, Houborg-, and Thomas- GRACE-indicators (Sec. 2) and potentially mask a drought. To this end, we select representative time series from each of the three synthetic grids of total water storage changes (TWSC) for



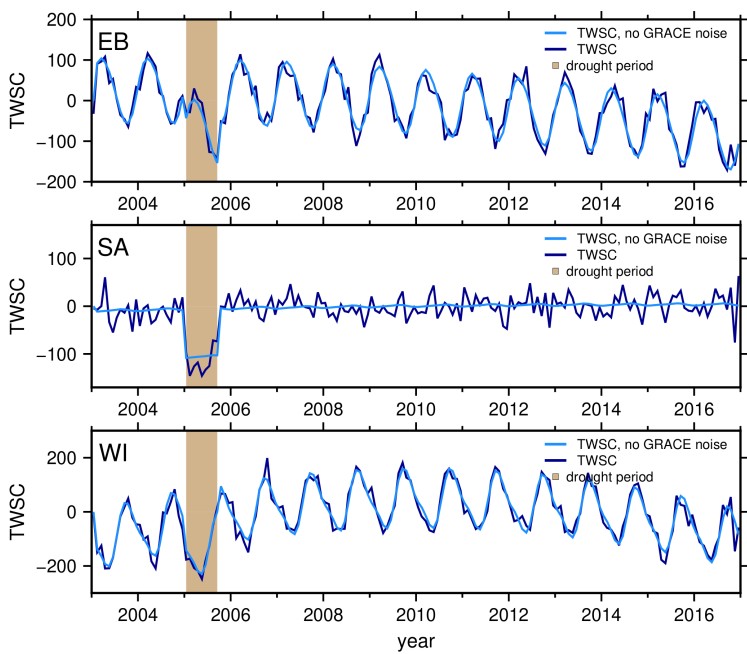

**Figure 5.** Synthetic TWSC [mm] without (light blue) and with spatial GRACE noise (dark blue) using average parameters for the clusters in East Brazil (EB), South Africa (SA), and West India (WI). Light brown shows the simulated drought period.

East Brazil (EB), South Africa (SA), and West India (WI), and apply the three methods. Since all results are based on TWSC, we refer to TWSC-DSIA, TWSC-DSID, TWSC-DIA and TWSC-DID as DSIA, DSID, DIA and DID.

We first assess the temporal characteristics of the Zhao-method (Sec. 2.1). Figure 6 (left) shows time series for the DSI and DSIA (with 3, 6, 12 or 24 months accumulated TWSC). It is obvious that trend and acceleration propagate into both DSI and
5 DSIA (see East Brazil and West India). Resulting indicator values e.g. for the years 2015 and 2016 are lower as compared to a small trend (South Africa) and this may lead to misinterpretations because a severe to mild drought is identified (-2 to -0.5) while none is actually simulated. In contrast, the actual simulated drought in 2005 is only identified as a moderate drought (values up to -1.0) for EB.

In the presence of a small trend (5.0 mm/year) and acceleration (-0.38 mm/year$^2$, Tab. 3, SA), we do identify exceptional
drought (Fig. 6 DSIA for South Africa). This shows that the drought strength that we chose does indeed would lead to a correct identification of exceptional drought in case no masking occurs (but in the presence of GRACE noise), so at this point we can determine that exceptional drought represents the 'true' drought severity class. As expected, a trend and/or an acceleration signal that are frequently observed in GRACE analyses can lead to misinterpretations in the indicators. However, the influence of the trend or acceleration also depends on the timing of the drought period within the analysis window. For example, assuming
we simulate the time series with the same trend or acceleration but the drought would occur in 2014, the drought detection





would not have been as much influenced. Therefore, we decided to set up an additional experiment and discuss the influence of different trend strengths for the drought detection (Sec. 4.3).

The analysis reveals that DSI and DSIA indicators are sensitive with respect to trends, while they are less sensitive to the annual and semi-annual signal. The seasonal signal is clearly dampened (compare e.g. Fig. 5 and the DSIA in Fig. 6). This is

caused by removing the climatology within the Zhao-method (Eq. 8). Comparing DSIA3, DSIA6, DSIA12, and DSIA24, e.g. for East Bazil, suggests that with longer accumulation period, indicator time series are increasingly smoothed and less severe droughts are identified (Fig. 6, left). Furthermore, the drought period appears shifted in time and its duration is prolonged. This can lead to missing a drought identification if a trend or an acceleration is contained in the analyzed timeseries, for example for the 24 months DSIA for East Brazil. We find that all DSIA are able to unambiguously detect a drought close to 2005 assuming

that neither trend nor acceleration is apparent (Fig. 6 DSIA for South Africa). In particular, the 3 and 6 months DSIA identify the drought close to 2005 for South Africa, and its computation appears to dampen the temporal noise that is present in the DSI.

In contrast we find that the 3, 6, 12, or 24 months TWSC-differencing DSID exhibit stronger temporal noise as compared to the DSIA and the DSI. This can be seen in the light of Eq. (2) - these indicators are closer to meteorological indicators and

thus do not inherit the integrating property of TWSC. The DSID does neither propagate a trend nor acceleration, annual signal or semi-annual signal. All DSID and DSID time series, for example for East Brazil (Fig 6, right), show a strong negative peak within the drought period, but this peak does not cover the entire drought period for the 3, and 6 months differenced DSID. The negative peak within the drought period is always followed by a strong positive peak, when we consider Eq. 2 this lends to the interpretation that a pronounced drought period is normally followed by a very wet event to return to 'normal' water

storage condition. Despite higher noise and the positive peak and contrary to the DSIA, all DSID (DSID3, DSID6, DSID12, and DSID24) correctly identify the drought within 2005 to be exceptional dry for East Brazil and South Africa. All different DSID time series for WI identify at least a moderate drought.

Analysis of the Houborg-method shows a broadly similar behaviour as compared to the Zhao-method: The sensitivity of drought detection to an included trend or acceleration depends on the indicators type. Using the DIA we can confirm the large

influence of the trend or acceleration on the indicator value, which is not the case for DID (e.g. Fig. 7 DIA and DID for East Brazil). Annual and semi-annual water storage signals are all considerably weakened in the Houborg-method because they are effectively removed when computing the empirical distribution for each month of the year. Differences to the Zhao-method appear when comparing more general properties, e.g. we find that DI is more noisy and the range of output values is restricted to about 7 % to 100 % (Fig. 7). This restriction is caused by the length of the time series, e.g. assuming we strive to identify

an event with exceptional dry values ($\leq 2\%$), we would need at least 50 years of monthly observations. Yet, with GRACE we only have about 14 years of good monthly observations, so the simulation was also restricted to this period. If we then take the driest value that might occur only once, we can compute the minimum value of DI to be 7.14 %. Hence the detection of exceptional or extreme drought is not possible when referring to the duration of the GRACE TWSC time series. As mentioned in Sec. 2.2, Houborg et al. (2012) applied a bias correction to the empirical CDF to mitigate this restriction. We do not follow

Houborg's approach here in order to focus on realistic observation availability instead of the availability of model outputs.

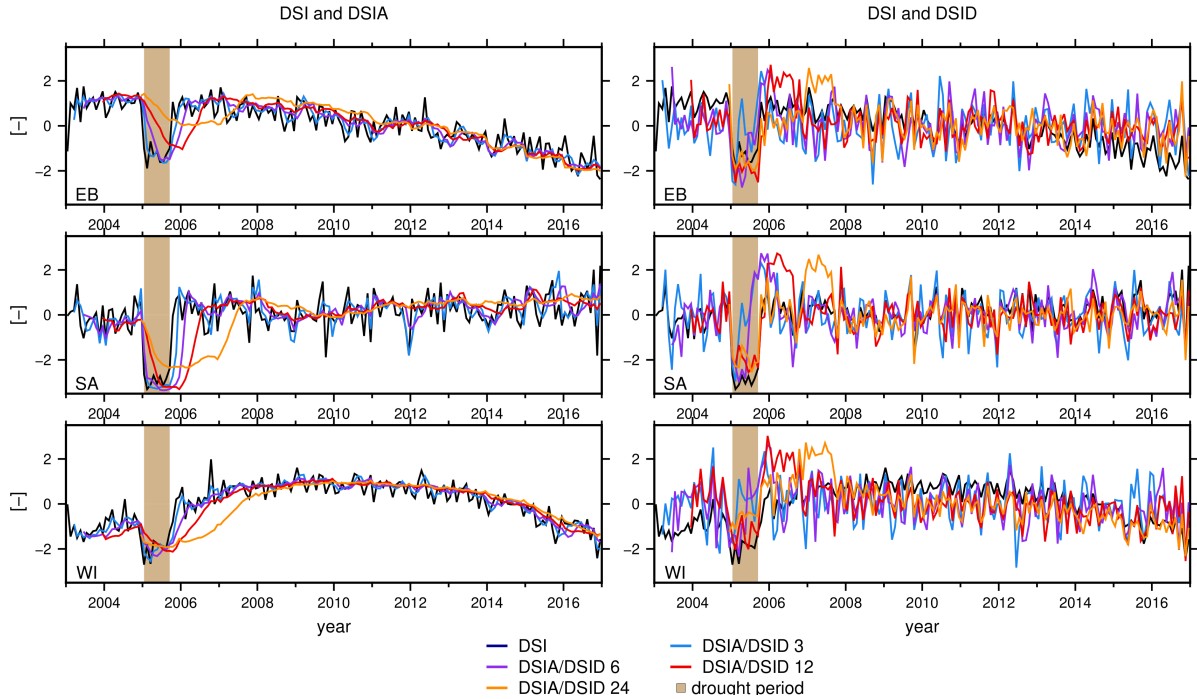

**Figure 6.** A representative example of the synthetic DSI, DSIA , DSID [-] for the East Brazil (EB), South Africa (SA), and West India (WI) cluster over the periods of 3, 6, 12, and 24 months. Light brown shows the synthetic constructed drought period.

Applying the Thomas-method to simulated GRACE TWSC results in magnitude, duration and severity of drought, which we show in Fig. 8 for the EB region. We find that the linear trend and acceleration propagate into the magnitude (Fig. 8, top) using TWSC deficits with climatology removed (blue, Eq. 6) instead of TWSC deficits with removed trends (linear and time-varying) and seasonality (red, Eq. 18). When using non-climatological TWSC (blue), we identify a strong deficit in 2015 and 2016 (Fig.

5   8, top) which suggests a duration of up to 28 months (Fig. 8, center) and a severity of about -2500 mm months (Fig. 8, bottom). Using the detrended and deseasoned TWSC (red), drought is mainly detected in the 'true' drought period (2005) and not at the end of the time series. Thus we conclude that a trend or acceleration indeed modifies the drought detection.

Results so far were derived by imposing a minimum duration of 3 months (blue and red). When moving to a minimum duration of 6 consecutive months (green, Fig. 8, middle and bottom) we find this would lead to a decrease in identified severity

10   by half, and the beginning of the drought period shifts 3 months in time. This is in line with Thomas et al. (2014). The same findings are made for South Africa and West India.

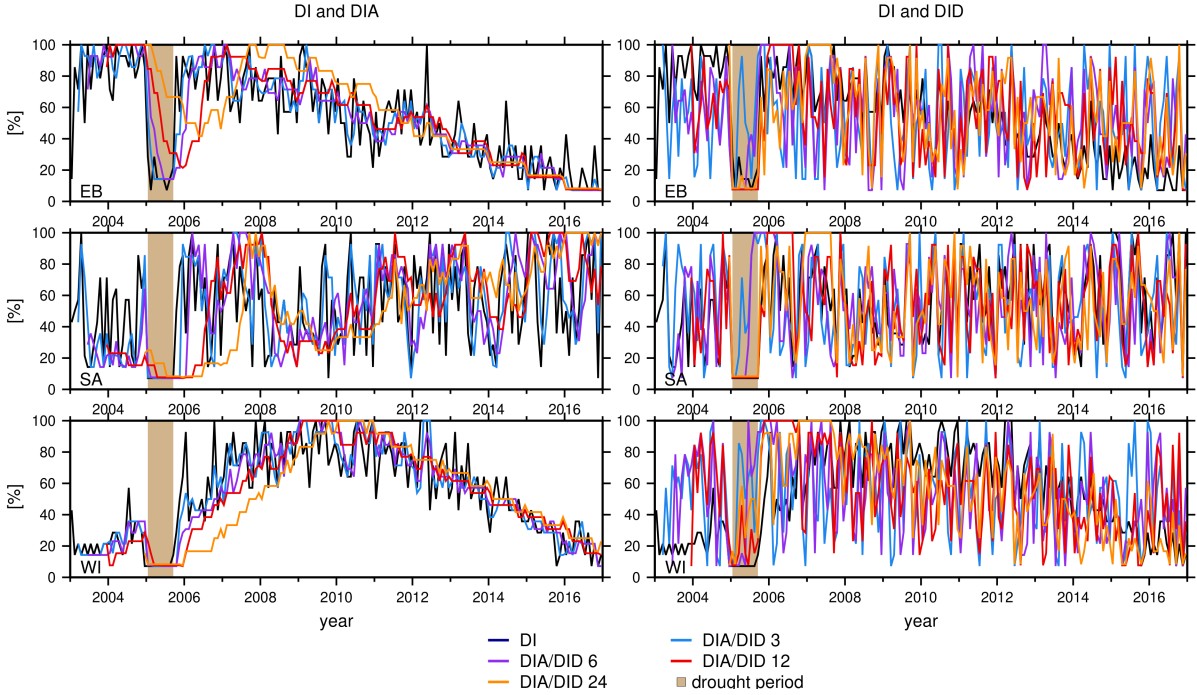

**Figure 7.** A representative example of the synthetic DI, DIA , DID [%] for the East Brazil (EB), South Africa (SA), and West India (WI) cluster over the periods of 3, 6, 12, and 24 months. Light brown shows the synthetic constructed drought period.

## 4.2 Synthetic TWSC: effect of spatially correlated GRACE errors

Here, we investigate how robust the Zhao-, Houborg- and Thomas-indicators are with respect to the spatially correlated and time-variable GRACE errors. However, any analysis must take into account that GRACE results cannot be evaluated directly at grid resolution.

In our first analysis, indicator based on (synthetic) TWSC grids are thus spatially averaged through two different methods (Sec. 3.1). We find that regional-scale DSI, DI indicators as well as the outputs derived by the Thomas-method for South Africa computed from 1) averaging TWSC first (darkblue Fig. 9) is indeed different to the 2) averaging indicators computed at grid scale from TWSC (lightblue, Fig. 9). These differences can be explained by the inherent non-linearity of the indicators. Since the synthetic data have been constructed from the same constants, trends, seasonal signal, temporal correlations, and drought

signal, we isolate the effect of GRACE noise on regional-scale indicators here. Outside of the drought period we conclude that the sequence how we spatially average causes larger differences for DI as compared to DSI: for South Africa, the range of averaged DI is about 7 - 100 % while the range of the DI of averaged TWSC is about 7 - 80 %. Within the drought period the DI exhibits little differences between both averaging methods. The DSI from averaged TWSC does suggest a weaker severity in the drought period compared to averaged DSI. In this case, both indicator averages identify the same (exceptional) drought

severity class. Yet we find that for DSI and DI the identification of drought severity is not sensitive to the choice of the averaging

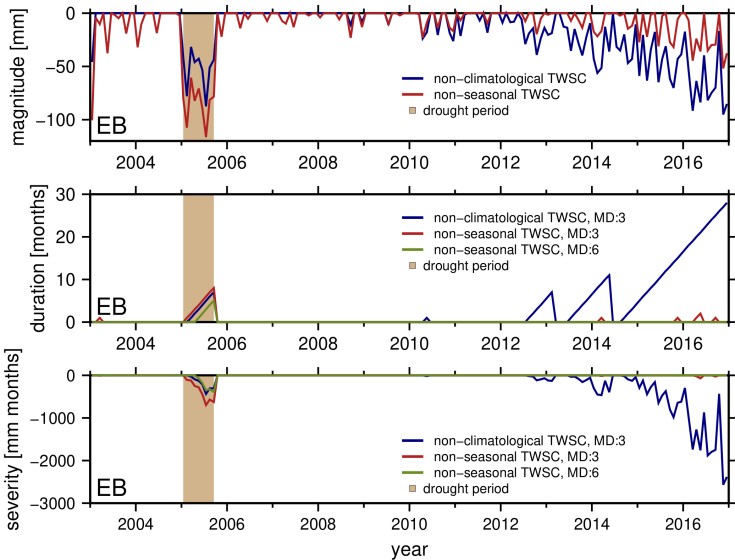

**Figure 8.** Drought magnitude [mm], duration [mo] and severity [mm·months] for the cluster of East Brazil (EB) using TWSC with removed climatology (dark blue) and TWSC with removed trend and seasonal signal (red). The minimum duration (MD) is set to 3 months (blue and red) or 6 months (green). Light brown shows the synthetic constructed drought period.

method for this cluster. However, for other cases differences can be more significant, which might lead to misinterpretation (e.g. February and April 2005 for the DI East Brazil, Fig. 9). For the Thomas-method, we cannot distinguish which result is more significant, since we have no comparable 'true' severity amount for that indicator.

To determine the influence of the GRACE-specific spatial noise on the detected drought severity, a second analysis is applied.
This analysis computes the share of area for each time step, for which a given drought severity class is identified (Fig. 10). Since different grid cells for one time step only differ in their spatial noise, it is important to understand that identifying more than one severity class is directly related to the noise. Only one class of drought would be detected for one epoch, assuming the grid cells have no or exactly the same noise. For example, we identify all classes of droughts (abnormal to exceptional) in December 2015 by using DSI for the East Brazil cluster (Fig. 10, top left). Thus, the spatial noise has a large influence on the
drought detection. To establish which indicator is mostly affected, the indicators are compared with each other.

We note that large differences are found between the DSI, the 6 months accumulated DSIA, and the 6 months differenced DSID within the given drought period for the East Brazil region (Fig. 10, left). All three indicators manage to identify the drought, but with different duration and percentage of affected area. The DSI shows exceptional drought within the drought period with a maximum of 38 % of the grid cells, i.e. it does not detect exceptional drought in all grid cells. On the contrary, the
15 DSIA does not detect exceptional drought in any grid cell. Apparently, this indicator misses the exceptional dry event because of the included trend and acceleration.



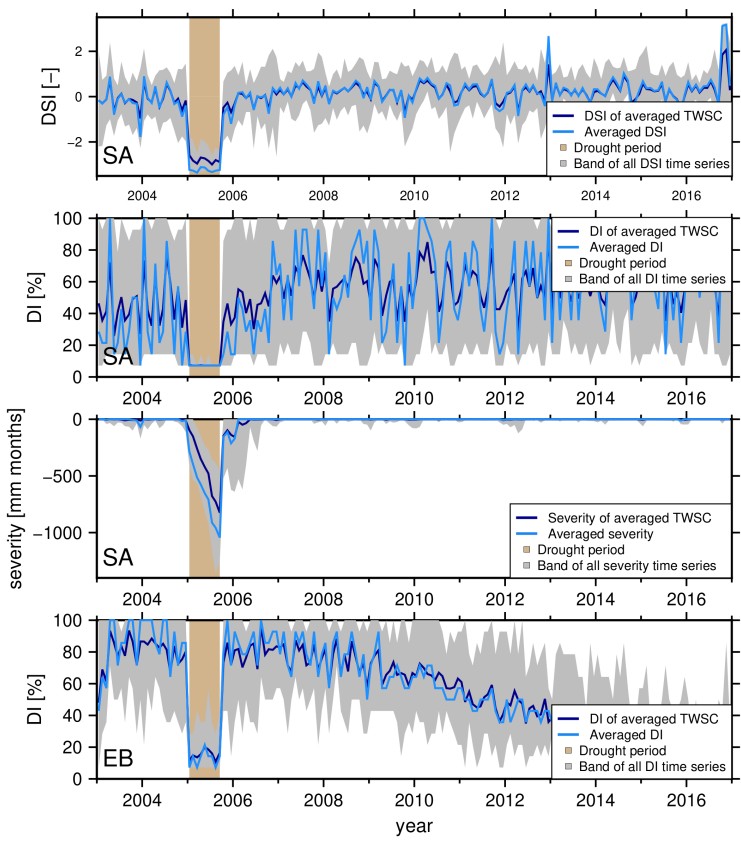

**Figure 9.** DSI and DI average in South Africa (SA, top and top center), severity average for the Thomas-method (SA, bottom center) and DI average in East Brazil (EB, bottom) by applying two different methods: the average of the indicators for all grids (light blue) and the indicators of averaged TWSC (dark blue). The grey shaded area represents the bandwidth for all grids. Light brown shows the synthetic constructed drought period.

When comparing DSIA of East Brazil to the DSIA of South Africa (Fig. 10, center), we find that DSIA is able to detect the drought strength correctly when there is a small trend or acceleration present. However, DSIA appears more robust against spatial noise, since it identifies (at least) severe drought in more than 90 % of grid cells, while the DSI indicator identifies only about 60 %. As described in Sec. 4.1, longer accumulation periods lead to smoother and thus more robust indicators. We find that the DSID is more successful in detecting exceptional drought: more than 60 % of the DSID grid cells show exceptional drought, but the indicator appears more noisy than the DSIA. Finally as what regards the drought duration, we find that only DSI detects the 'true' period correctly. When identified via DSIA, the duration appears longer and when identified in DSID, the period was found shorter as compared to the 'true' drought period.





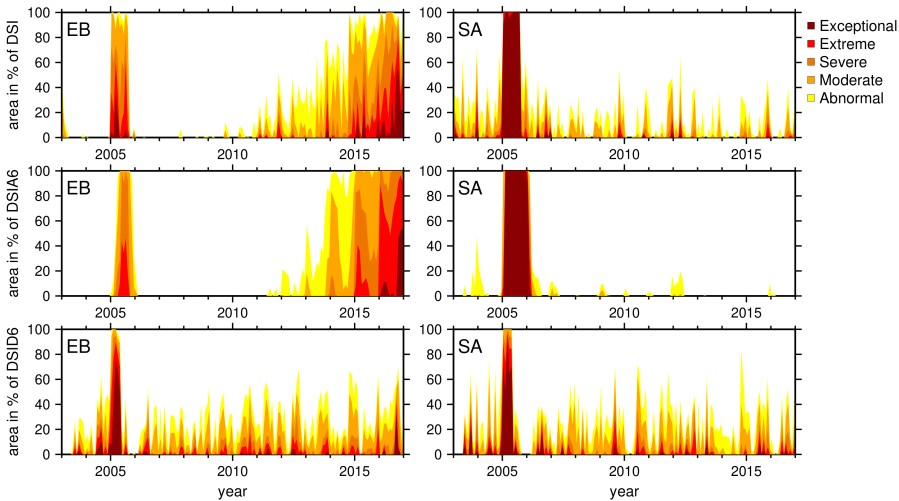

**Figure 10.** Drought affected area of the DSI, DSIA, and DSID [%] considering the different drought severity classes within the clusters of East Brazil (EB) and South Africa (SA).

Overall, we find that the different indicators DSI, DSIA or DSID all come with advantages and disadvantages regarding the presence of spatial and temporal noise. The same findings were made for the indicators of the Houborg-method (results not shown). This analysis is not applied to the Thomas-method, because the method does not refer to severity classes (Sec. 2.3).

### 4.3 Synthetic TWSC: experiments with variable trend, drought duration and severity

Two experiments were additionally constructed to examine the influence of trends and drought parameters on the indicator skills. First, we consider how strong a linear trend in total water storage must be to mask drought in the indicators. For this, we test different trends from -10 mm/year to 10 mm per year for DSI, DSIA, DI, DIA and the Thomas-method in the West India region (since these indicators were identified as being affected by trends, Sec. 4.1). No acceleration is included for these tests. We find that trends between -1 and 1 mm per year cause no influence on all indicators, while differences start to appear when

simulating a trend higher than 2 mm per year. This propagates into DSI, DSIA, DI and DIA indicators but did not affect the drought period.

What would be the largest trend magnitude that does not affect the correct detection of drought duration and drought severity, and how can we verify this? An obvious influence within the drought period in 2005 is found when simulating a trend of -6 mm or lower per year. It is important at this point to understand that there is a relation between the timing of the drought and

15 the sign of the trend, i.e. a positive or a negative trend. Assuming that a positive trend exists and the drought occurs closer to the end of the time series, the trend may lead to a drought that is identified as more dry than the actual drought. But if the trend is negative, the drought is identified more easily.





Other factors, e.g. the length of the time series, have an influence on the masking by the trend and, as a result, affect drought detection. The longer the input time series, the more sensitive is the drought detection to the trend. At the same time, the magnitude of the trend needs to be considered relative to the variability or range of the TWSC. E.g. -6 mm per year trend has a larger influence on the drought detection assuming the range of TWSC being -50 to 50 mm as compared to -200 to 200 mm.

As a reference, the synthetic time series for West India, without any trend or acceleration signal, ranges from about -335 to 76 mm. So, deriving a general quantity for these dependencies is difficult.

In a second experiment, we assess which input drought duration and magnitude would at least be visually recognized in the indicators. We choose 3, 6, 9, 12, and 24 months for the simulated duration and -40 mm, -60 mm, -80 mm, -100 mm -120 mm for the drought magnitude, and apply both the Zhao- and the Houborg-method. We compare the changes for one indicator

time series for the East Brazil region. The drought always begins in January 2005 for the first tests. In general, we found that the identification of the severity class is less sensitive to changes in the drought duration, since a drought duration of 3, 6, 9, 12 and 24 months mostly results in equal drought severity classes for example for a drought magnitude of 120 mm. Thus, we concentrate our analysis on changes in drought magnitude.

The severity class with the strongest drought type (i.e. exceptional drought) is only classified by the Zhao- and Houborg-

method for East Brazil when using a drought magnitude of -120 mm; this is related to the trend and acceleration signal contained in the simulated TWSC and was already found in Sec. 4.1. For the Zhao-method, extreme drought is identified when simulating a drought magnitude of at least -100 mm, while only severe and moderate drought is identified when simulating a magnitude of -80 mm and -60mm. The Houborg-method fails to identify extreme and exceptional drought, as described in Sec. 4.1. Thus, a magnitude of -80 mm in severe drought all applied drought periods (3 to 24 months), while a magnitude of

-60 mm leads to moderate dry events and a magnitude of -40 mm to abnormal dry events. We find that the both methods are not able to clearly detect a drought that has a magnitude of -40 mm or higher, if the duration is between 3 and 24 months. This experiment supports our findings in Sec. 3.2.

## 4.4 Application to real GRACE data: South Africa droughts

For South Africa, droughts are a recurrent climate phenomenon related to climatic conditions. The complex rainfall regime

led to multiple extents of drought events in the past, for example to a strong drought in 1983 (e.g. Rouault and Richard, 2003; Vogel et al., 2010; Malherbe et al., 2016). The droughts appeared in varying climate regions at different timing of the year and with a different severity. Since 1960, many of them were linked to El Niño (e.g. Rouault and Richard, 2003; Malherbe et al., 2016).

Based on the simulation results, we chose the 6 months accumulated DSIA to identify droughts for (the administrative area

of) South Africa (*GADM*, 2018) in retrospective in the GRACE total water storage data. DSIA has proven to be more robust with respect to the peculiar, GRACE-typical spatial and temporal noise as compared to the other tested indicators (Sec. 4.2 and 4.1).

GRACE-DSIA6 suggests two drought periods, from mid of 2003 to mid of 2006 and from 2015 to 2016 (Fig. 11). The first drought event is identified to affect at least 70 % of the area of South Africa. While 2003 was indeed a year of abnormal to



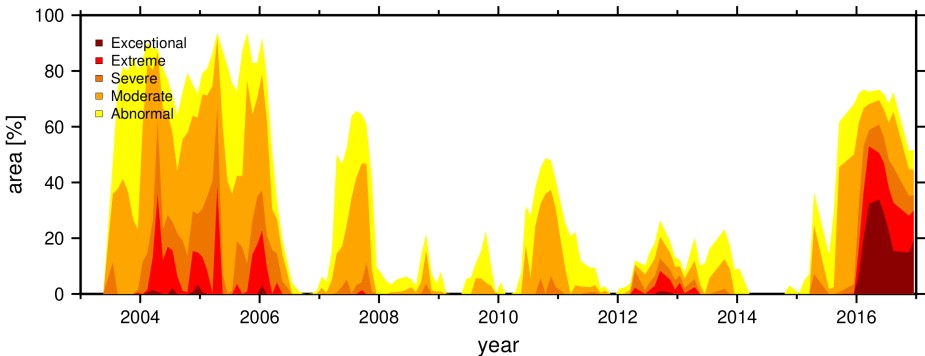

**Figure 11.** Percentage of drought affected area of the 6 months DSIA[-] considering the different drought severity classes. Application on real GRACE-TWSC over South Africa from 2003 to 2016.

severe dry conditions, in 2004 until mid of 2006 also extreme drought occurred. Figure 11 reveals that a small area (about 7976 km$^2$, close to Lesotho) experienced even exceptional drought in 2004. This period is confirmed by The Emergency Events Database (EM-DAT , 2018) recording a drought event in 2004, see e.g. Masih et al. (2014). Extreme drought in 2004 mainly occurred in the Central and South East of South Africa; this is exemplarily shown for April 2004 in Fig. 12a. Another

confirmation is found in Malherbe et al. (2016), who identified a drought period from 2003 to 2007 by using the SPI.

The second drought in 2015 and 2016 is identified to have affected less area (about 50 to 70 %, Fig. 11), but it is perceived as more intense than the 2003 to 2006 drought. Based on GRACE and the DSIA6, we conclude that at least 30 % of South Africa were affected by extreme drought and about 20 % experienced an exceptional drought in 2016. The 2016 drought occurred in the North Eastern part of South Africa (Fig 12b). For comparison, the EM-DAT database also listed 2015 as drought event

but not 2016. We speculate that the differences are due to the drought criteria of the EM-DAT database (disasters are included when, for example, 10 or more people died or 100 or more people were affected). However, the EM-DAT database lists 2016 as a year of extreme temperature, which might be related to our detected drought. Furthermore, we can confirm the 2015/2016 drought by a lower maximum precipitation in these years than in other years (about 65 mm) and by meteorological indicators indicating severe to extreme drought (SPI, Standardized Precipitation Evapotranspiration Index (Vincente-Serrano et al., 2010),

and Weighted Anomaly Standardized Index (Lyon and Barnston, 2015)).

## 5   Discussion

The framework developed in this study enables us to simulate GRACE-TSWC data with realistic signal and noise properties, and thus to assess the skills of GRACE drought indicators in a controlled environment with known 'truth'. This will be extended to GRACE-FO in the near future. GRACE studies have been often based on simplified noise models (e.g. Zaitchik et al., 2008;

Girotto et al., 2016); however it is important to account for realistic error and signal correlation, in particular for drought studies where one will push the limits of GRACE spatial resolution.

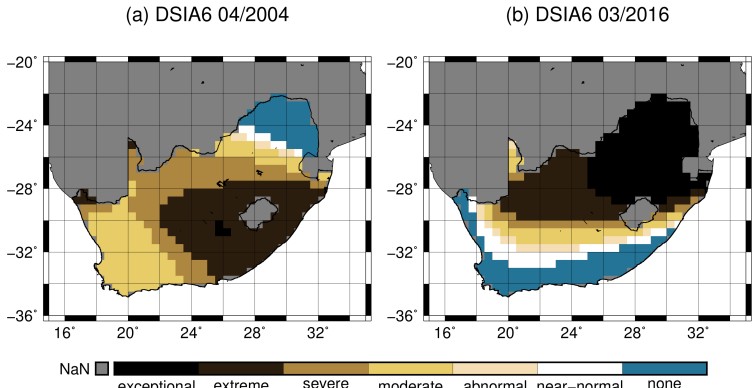

**Figure 12.** DSIA6 [-] for real GRACE-TWSC within South Africa (black line, *GADM* (2018)) for (a) April 2004 and (b) March 2016.

However, identifying a drought signal from real GRACE-TWSC is indeed challenging since we do not know in advance how the signature of a drought looks like; a parametric drought model does not yet exist and our experiment (Sec. 3.2) to extract such a model from TWSC data and known droughts did not lead to conclusive results. Still we believe that this first – to our knowledge – approach identified a similar systematic behaviour of different drought periods, although based on a small number

of drought periods, and should be pursued further. Based on literature and our own experiments (Sec. 4.3) we chose to define our 'box'-like GRACE drought model as an immediate and constant water storage deficit.

When analysing the Zhao-, Houborg- and Thomas-methods, we find that trends and accelerations in GRACE water storage maps tend to bias the DSI, DI and the Thomas-indicator that uses non-climatological TWSC, but also for the DSIA and DIA which use accumulated TWSC. Indicators DSID and DID, which utilize time-differenced TWSC, were not found biased by

trends and accelerations; the same goes for the Thomas-method when based on detrended and deseasoned TWSC. When we did not simulate a trend, all indicators were able to detect drought, but they identified different timing, duration, and strength. This suggests removing the trend in GRACE data first, but this must be done with care, since it can also influence the detection of, for example, long-term droughts. The same is true for removing the trend and seasonal signal prior of applying the Thomas-method, although in this study we found that the removal of these signals simplified the correct drought detection (Sec. 4.1).

An experiment was then set up to understand the influence of the trend on the detected drought duration and severity. Several factors play a role here, e.g. the length of the time series, the TWSC range in relation to the trend magnitude, and the sign of the trend. We found that providing a general rule appears nearly impossible.

As expected, we find time-series for the modified time-differencing GRACE indicators DSID and DID as much noisier when compared to the time-accumulating indicators DSIA and DIA; this can be linked to precipitation (Sec. 2) driving total

water storage. The drought period was identified to be shorter than the 'true' simulated drought period for e.g. for DSID3 and DSID6. After these drought periods, strongly wet periods were detected. In the applications, we suggest a direct comparison of the DSID and meteorological indicators in particular for confirming or rejecting drought duration and the following wet periods.





On the contrary, computing accumulated indicators implies a temporal smoothing and the drought period will appear lagged in time, albeit for accumulation periods of 3 and 6 months the lag was found insignificant. DSIA and DIA are thus more robust against temporal and spatial GRACE noise as compared to DSID and DID, and again we would suggest 3 or 6 months accumulation periods. In general, we found the Zhao- and Thomas-indicators performing better in detecting the correct drought

strength than the Houborg-method, at least seen the limited duration of the GRACE time series that we have at the time of writing.

By simulating the effect of spatial noise on drought detection, we found that some indicators appear less robust. Analysis of the percentage of drought affected area showed that the GRACE spatial noise limits the correct drought detection. Again, the DSIA was identified to be more robust as compared to DSI and DSID - it was the only indicator that identified exceptional

drought in nearly all grid cells. A second experiment was applied to examine, if the influence of the spatial noise can be reduced by using spatial averages. We found that spatially averaging DSI and DI appears less robust against the spatial noise compared to computing the indicator of averaged TWSC. At this point we therefore suggest to compute the indicator from spatially averaged TWSC. Since the DI showed stronger difference between both averaging methods than the DSI, we conclude that the DI is generally less robust against spatial noise than the DSI. In our real-data case study, due to these findings, the DSIA6 was

then applied to GRACE-TWSC, and it identified two drought periods: mid 2003 to mid 2006 in Central and South East and 2015 to 2016 in North East of South Africa.

## 6   Conclusions and outlook

A framework has been developed that enables understanding the masking of drought signals when applying the Zhao et al. (2017), Houborg et al. (2012) and Thomas et al. (2014) methods. Four new GRACE-based indicators were derived and tested;

these are modifications of the above mentioned approaches and work with time-accumulated and -differenced GRACE data. We found that indeed most indicators were mainly sensitive to water storage trends and to the GRACE-typical spatial noise.

Among these various indicators, we identified the DSIA6 as in particular well-performing, i.e. less sensitive to GRACE noise and with good skills in identifying the correct severity of drought at least in absence of trends. However, the choice of the indicator should always be made in the light of the application.

We see ample possibilities to extend our framework. Future work should focus on better defining the begin and end of a drought and developing a signature for TWSC drought. One will also consider other observables in the simulation such as e.g. groundwater, which can be derived from GRACE and by removing other storage contributions from direct modelling or through data assimilation.

In the GRACE community, efforts are currently being made to 'bridge' the GRACE timeseries to the begin of the GRACE-

FO data period (e.g. Jäggi et al., 2016; Lück et al., 2018). These gap-filling data will inevitably have much higher noise and spatial correlations that may be very different from GRACE data, and drought detection skills should be investigated through simulation first. On the contrary, GRACE-FO is supposed to provide more precise measurements, and thus less influence of spatial noise on the drought detection may be expected. The combination of GRACE-FO data and a thorough understanding

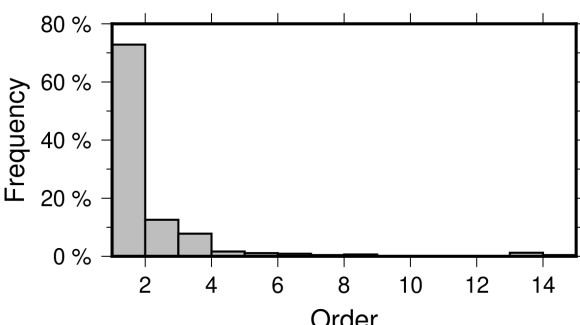

**Figure A1.** Histogram of the optimal order of an AR model for global detrended and deseasoned GRACE-TWSC on land grids.

and 'tuning' of GRACE drought identification methods, possibly through this framework, might then enable us to identify water storage droughts more precisely.

## Appendix A:  AR model coefficients computations

To extract temporal correlations from the GRACE total water storage changes (TWSC) we apply an autoregressive(AR) model, which is described by

$$X(t) = \phi_1 X(t-1) + ... + \phi_p X(t-p) + \epsilon_t, \tag{A1}$$

where $X$ represents the observed process at time $t$, $p$ is the model order, $\phi$ are the correlation parameters, and $\epsilon$ is a white noise process (Akaike, 1969). Here, detrended and deseasoned TWSC are used as the observed process $X(t)$, because the remaining residuals contain interannual and subseasonal signal as the drought information, which we want to extract with this approach. The approach is then applied for different model orders. The optimal order of the AR-model is adjusted by means of the information criteria, for example the Akaike information criterion (AIC), and the Bayes information criterion (BIC). Then, by using the optimal order, the AR-model coefficients $\phi$, which represent the temporal correlations, can be computed using a least squares adjustment.

The results for the optimal order of interannual and subseasonal TWSC is shown in Fig. A1. The most of the global land grids of detrended and deseasoned TWSC shows an optimal order of 1 (about 70%).

## Appendix B:  Eigen value decomposition

The decomposition of the variance-covariance matrix $\Sigma$ by using Cholesky decomposition fails, when $\Sigma$ is positive semi definite. To still be able to decompose the matrix, we can use eigen value decomposition, but this is accompanied by a loss of information due to the rank deficiency. The decomposition is then examined by $\Sigma = UDU^T$, where $U$ is a matrix with the eigenvectors of $\Sigma$ in each column and $D$ is a diagonalmatrix of the eigenvalues. In this case, a decomposed matrix can be



related to $R^T$ introduced in Sec 3.1. $R^T$ can be computed by $U\sqrt{D}$. In Sec. 3.1, we multiply $R^T$ with a normal distributed noise time series of the same length as the rows of $\Sigma$. In this case, the number of normal distributed noise time series $n$ is then replaced by the rank of $\Sigma$.

*Author contributions.*  HG, OE, and JK designed all computations and HG carried them out. HG prepared the manuscript with contributions from OE and JK.

*Competing interests.*  The authors declare that they have no conflict of interest.

*Acknowledgements.*  We acknowledge the funding from the Federal Ministry of Education and Research in Germany (BMBF) for the 'GlobeDrought' project through through its funding measure Global Resource Water (GRoW) (FKZ 02WGR1457A).



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
