# Peer review of "A framework for deriving drought indicators from GRACE"

_Hydrology and Earth System Sciences, 2019_

## Referee Comment (RC1) · Anonymous Referee #1 · 20 Aug 2019

The authors developed a framework for deriving synthetic terrestrial water storage change (TWSC) from the GRACE observations for computing drought indicators. The synthetic TWSC enables comparisons of existing drought indicator methods and analyses of the influence of GRACE trend and noise on the drought detection. I think that the topic is important for further understanding of hydrological drought and the findings are valuable, shedding lights to characteristics of different drought identification methods. The manuscript is fairly well written; however, I have some concerns and need some clarifications. Thus, I recommend major revision and the specific comments are listed below.

1. The authors chose three existing indicators of Zhao et al (2017), Houborg et al (2012), and Thomas et al (2014) methods because they are based on the monthly

[Figure]

GRACE data. The comparison of the methods is interesting, but I don't think this framework is a fair ground for evaluating their skills, especially for the Houborg-method. As I understand it, the CDF (which is the basis for the percentile computation) is based on the historical simulation of 1948-2010. This analysis focuses on the GRACE period of 2002-2016 and it mentions about disregarding the bias correction. Also for Zhao-method, a bias correction is not applied. I understand that direct comparison of these indicators are not possible as Houborg is regional, but the indicators in their final product form (as opposed to the method concepts) may be able to detect the drought that were missed in this study. 2. Is the GRACE-specific noise dependent on the instrument or the solution? As it is an important term and needs to be characterized well, I am wondering if it would be different when using different GRACE-TWSC solutions such as mascon solutions from JPL or CSR. Is the same approach (equation 21) applicable to other GRACE data? What is the grid size of the TU GRAZ data (0.5 degree)? 3. [Page4;Ln9-18] The equations 1-3 are not referred later in the text. I agree that TWSC corresponds to precipitation anomaly accumulation in many cases, but it does not seem to tie in with the rest of the discussions. 4. [Page9; Ln5-7] Identifying regional clusters seems very important and I wonder where else clusters are located. 5. [Page9; Ln16-17] It would be helpful to present the list of droughts included in step 3, in a table or supplement. 6. [Page10,Ln20] It was not clear to me if the study addressed the last point of this framework's benefit "…identify the most suitable indicator for a specific application". 7. [Figure 5] I understand that the purpose of this figure is to show the importance of spatial GRACE noise, especially in SA. However, the TWSC time series for EB and WI have low TWSC amplitudes that are equally as low as that of during the simulated drought period in later 2016 (EB) and 2003/2004 (WI). Can you add an explanation to what distinguishes 2005 from these low TWSC? 8. [Section 4.4] I am a bit confused how Figure 11 using real GRACE data is very different from the corresponding figure 10 center, right using the synthetic TWSC. Related to point 5 above, the synthetic data can detect only 2005 drought by design? 9. [Page21; Ln20] It is not clear what "simplified noise models" mean. Please elaborate. 10. [Page22; Ln10-11] I

do not follow "when we did not simulate a trend". When did you? 11. [Page23; Ln10] It will be helpful to name the four new indicators (or refer to equations).

Minor edits: [Equation 19] dot typo? [Table 3] What are the two values for Annual and Semi-annual? [Page14;Ln16] DSID appears twice. [Page15;Ln1] This sentence is incomplete. [Figure 6] DSI appears as black line in the plots while legend for DSI is blue. [Page21;Ln7] GRACE and the DSIA6 -> GRACE DSIA6?

---

## Referee Comment (RC2) · Anonymous Referee #2 · 13 Sep 2019

Review of HESS-2019-268 "A framework for deriving drought indicators from GRACE" by Helena Gerdener; Olga Engels; Jürgen Kusche

Recommendation- Major revision

General Comments- In this paper the authors developed a framework that potentially contributes to the understanding of how drought signals propagate through various GRACE drought indicators. By applying three methods (GRACE-based indicators), the authors assessed the skills of newly derived GRACE drought indicators under rather more controlled conditions. This work is significant, as the study is a considerable addition to the existing literature about drought identification methods. Also, the topic is within the scope of Hydrology and Earth System Sciences. Overall, the experimental design is clear, and for the most part, the authors' conclusions are supported by their

findings. However, I outline several general concerns, followed by a range of specific comments, which prevent me from recommending this manuscript for publication in its current form. I do hope though that the authors will be able to adequately address my comments and when that is done, this paper should be acceptable for publication.

1. The paper is relatively poorly written. There is a significant number of grammatical/syntactic errors that are present throughout the entire body of the manuscript. I specify several of these in the "Specific Comments" section below, but the authors need to thoroughly check the entire text, as similar or other mistakes may exist elsewhere. 2. Page 3 Line 14 "As can be expected, TWSC and 6 months SPI appear moderately similar (correlation 0.43), characterised by positive peaks e.g. at the beginning of 2004 and at the end of 2009, and negative peaks at the beginning of 2013. This motivates us to modify common GRACE indicators. . ." I find the evidence not supportive enough to safely conclude that this link/association between TWSC and SPI is always (or everywhere) the case. The authors should test this on several different regions characterized by varying hydro-climatic conditions. Making such conclusive statements using only one example is scientifically inaccurate. 3. More information is required for the cluster identification. How exactly were the three clusters determined? The authors also need to clearly specify their exact geographic locations. 4. The authors should provide more detailed information (characteristics) about specific droughts mentioned in their methodology section.

Specific Comments-

Abstract

"Thus, this study aims at a better understanding of how drought signals, in the presence of trends and GRACE-specific spatial noise, propagate through GRACE drought indicators": This phrase is perhaps the essence of the abstract; therefore it should be able to provide the necessary information on its own. The authors need to specify which trends they are referring to. Line 10 application-dependent Line 10 large differences Line 11 particularly Line 12 We show that trends and accelerations – what do the authors mean by "accelerations"?

Page 1 Line 17 affect the Line 18 replace "reach" with "range" Line 24 led

Page 2 Line 4 depends on the accumulation period considered - unclear Line 16 Much fewer Line 23 and the first data are expected Line 27 they found good agreement to net precipitation minus evaporation. - unclear Line 34 without utilizing external information – please specify

Page 3 Line 4 delete "e.g." Line 7 "smoothing" Line 17 What are "differencing periods"? Line 21 spatially averaged Line 26 will complete the paper

Page 4 Line 2 explore Line 10 more regularly

Page 8 Line 10 we construct Line 13 including the introduced (in Sec. 2.3) signal . . . Line 26 . . .following A et al. (2013). . ...is there something missing here?

Page 11 Line 8 drought onset and end Lines 10-14 these thresholds are rather arbitrarily made. It seems to me that a single value for the drought duration and magnitude should not be used for different hydrologic regimes.

Page 12 Line 5 inappropriate use of English for a scientific paper

Page 13 Line 10 delete "would"

Page 14 Line 17 for the 3, and 6 months differenced DSID

Page 20 Line 24 climatic phenomenon Line 24 delete "related to climatic conditions" as it is redundant

Page 21 Line 9 in the northeastern

Page 23 Line 22 particularly Line 25 the onset and end

Please also note the supplement to this comment:

https://www.hydrol-earth-syst-sci-discuss.net/hess-2019-268/hess-2019-268-RC2-supplement.pdf

---

## Editor Comment (EC1) · Bettina Schaefli (Editor) · 13 Sep 2019

Both reviewers agree that this is potentially an interesting paper for HESS but that it requires some additional details on methodological choices. I invite the authors to answer these comments in the public discussion before preparing the revised version.
* * *

---

## Author Comment (AC1) · 4 Oct 2019

**Author comment to Reviewers comment #1**

Helena Gerdener, Olga Engels and Jürgen Kusche

The author's answers are indicated in red color, as well as old text passages. New text passages are indicated in green color.

The authors developed a framework for deriving synthetic terrestrial water storage change (TWSC) from the GRACE observations for computing drought indicators. The synthetic TWSC enables comparisons of existing drought indicator methods and analyses of the influence of GRACE trend and noise on the drought detection. I think that the topic is important for further understanding of hydrological drought and the findings are valuable, shedding lights to characteristics of different drought identification methods. The manuscript is fairly well written; however, I have some concerns and need some clarifications. Thus, I recommend major revision and the specific comments are listed below.

Response:
Thank you very much for the time spent in reviewing the manuscript and for the really useful reviewer comments, which certainly helped us to improve the manuscript.

**Comment 1.**

The authors chose three existing indicators of Zhao et al (2017), Houborg et al (2012), and Thomas et al (2014) methods because they are based on the monthly GRACE data. The comparison of the methods is interesting, but I don't think this framework is a fair ground for evaluating their skills, especially for the Houborg-method. As I understand it, the CDF (which is the basis for the percentile computation) is based on the historical simulation of 1948-2010. This analysis focuses on the GRACE period of 2002 to 2016 and it mentions about disregarding the bias correction. Also for the Zhao-method, a bias correction is not applied. I understand that direct comparison of these indicators are not possible as Houborg is regional, but the indicators in their final product form (as opposed to the method concepts) may be able to detect the drought that were missed in this study.

Response:
Thanks, we understand the reviewer's point. However, we believe the primary aim of this paper is to provide a fair comparison of the indicators based on a synthetic environment, which is derived by computing simulated TWSC during the GRACE period. Within this controlled synthetic environment the fairest comparison for the Houborg-method is a comparison without using a bias correction. We agree with the reviewer that a bias correction would be appropriate when considering real observations of TWSC.

We believe that this issue would require further discussion which we cannot provide here. Our simulations confirm that without applying any correction, indeed the limited duration of the synthetic time series may render the computation of the biased indicator. Nonetheless, the bias correction as suggested in Houborg's paper would have to come from a cumulative distribution function (CDF) derived from long runs of hydrological models, and these are far from representing reality as new studies show (e.g. Scanlon et al., 2018).

On balance, as our main focus is on the synthetic environment, we prefer to keep our current indicator computation, but we modified a sentence in the description of the Houborg-method, to precisely state our focus.

Old text:
Here, we focus on a TWSC from GRACE only and, as explained in Sec. 2.1, we therefore disregard the bias correction.

New text:
Here, we focus on a simulated TWSC environment for the GRACE period only and, as explained in Sec. 2.1, we therefore disregard the bias correction.

**Comment 2.**

Is the GRACE-specific noise dependent on the instrument or the solution? As it is an important term and needs to be characterized well, I am wondering if it would be different when using different GRACE-TWSC solutions such as mascon solutions from JPL or CSR. Is the same approach (equation 21) applicable to other GRACE data? What is the grid size of the TU GRAZ data (0.5 degree)?

Response:
Even after 17 years of GRACE data, a full understanding of the GRACE noise characteristics, let alone of the individual sources, has not been reached. The noise in the GRACE solutions depends on the GRACE orbit configuration, on the instrument performance (which changed significantly over time due to technical issues such as the switch-off of the thermal stabilization of the accelerometers in 2010), of the realism of the background models, and on the data editing and estimation strategy itself which differs between institutions. One could either use a diagonal or a non-diagonal solution variance-covariance matrix to describe the noise model. By accounting for a non-diagonal solution variance-covariance matrix, the noise model accounts for latitudinal variation of noise levels dependency due to orbit convergence. However certain errors like the noise introduced by background model errors are difficult to know and, currently, there is no way of accounting for them. So the short answer is one would probably be able to work with the same error characterization for other GRACE solutions.

However, the use of the mascon solutions creates another difficult problem; the mascon solution exhibit a better S/N ratio as compared to the conventional solutions but this is to a large extent due to the fact that these solutions use constraints derived from geophysical models, and it would be difficult to characterize the biases introduced by these constraints.

Here, we use the TU GRAZ solutions that are provided in monthly geopotential coefficients (spherical harmonics, SH), this means the data is not given in the spatial domain. We then transform these coefficients by using spherical harmonic synthesis to monthly TWSC grids (here we use 0.5 degree grid). These grids inherit the native GRACE resolution which is somewhere about 300 km. The geopotential coefficients can also be derived by other processing centers, for example CSR, GFZ and JPL. Therefore, we could also apply our approach on these data.

The reviewer raises a very important point about the significance of a proper noise description. The SHs used to compute TWSC are provided along with corresponding standard deviations (ftp://ftp.tugraz.at/outgoing/ITSG/GRACE/ITSG-Grace2016/monthly/monthly_n60/). In the submitted version, we propagated this information to a grid, which led to a full variance-covariance matrix (used in Eq. 21) for the TWSC. Following the reviewer comment, we now use a full variance-covariance matrix (normal equations provided by TU GRAZ: ftp://ftp.tugraz.at/outgoing/ITSG/GRACE/ITSG-Grace2016/monthly/monthly_n90_normals/) of the SHs to estimate the full variance-covariance matrix of the TWSC. This procedure better represents the GRACE-specific noise, because the correlations between the SHs are taken into account. Thus, at the moment the full variance-covariance matrix of the SHs is in our opinion the best solution available to describe the GRACE-specific noise.

After this adjustment, some passages, values and figures have been slightly changed, but the interpretation of all derived results remained exactly the same. These changes are mentioned below under the section "Changes in the noise term".

**Comment 3.**

[Page4;Ln9-18] The equations 1-3 are not referred later in the text. I agree that TWSC corresponds to precipitation anomaly accumulation in many cases, but it does not seem to tie in with the rest of the discussions.

Response:
Following the reviewers comment, we reference the Eq. 1 that is used to better describe the relation of Eq. 2 (Page4;Ln16). The Eq. 2 and 3 are referred to accumulated (Page5;Ln21-24) and differenced (Page 5-6;Ln26-3) TWSC, correspondingly. For the sake of completeness and to avoid any misunderstandings regarding the connection between fluxes and storages, we would like to keep these equations.

**Comment 4.**

[Page9;Ln5-7] Identifying regional clusters seems very important and I wonder where else clusters are located.

Response:
We agree this needs an additional figure, which we added to the Appendix (B1) and adjusted the text of the manuscript correspondingly.

Old text1:
As a result of this procedure, we chose three clusters located in East Brazil (EB), South Africa (SA), and West India (WI), which were also affected by droughts in the past (e.g. Parthasarathy et al., 1987; Rouault and Richard, 2003; Coelho et al., 2016).

New text1:
As a result of this procedure, we identified three clusters located in East Brazil (EB), South Africa (SA), and West India (WI), which were indeed affected by droughts in the past (e.g. Parthasarathy et al., 1987; Rouault and Richard, 2003; Coelho et al., 2016). Location and shape of the three chosen clusters are shown in Fig. 3 and a global map of all clusters is provided in Fig. B1.

[Figure]

Figure B1. Clusters based on Expectation Maximization (EM) clustering applied to the global autoregressive model (AR) coefficients.

**Comment 5.**

[Page9;Ln16-17] It would be helpful to present the list of droughts included in step 3, in a table or supplement.

Response:
Thanks, for the suggestion. We added a table to show the considered TWSC period for the corresponding drought periods.

Old text:
Searching for drought duration and magnitude (step 3) led to four droughts seen in GRACE-TWSC: The 2005 and 2010 droughts in the Amazon (e.g. Chen et al., 2009; Espinoza et al., 2011), the 2011 drought in Texas (e.g. Long et al., 2013), and the 2003 drought in Europe (e.g. Seitz et al., 2008).

New text:
Performing literature research for drought duration and magnitude (step 3) led to four droughts seen in GRACE-TWSC (Tab. 4): The 2005 and 2010 droughts in the Amazon (e.g. Chen et al., 2009; Espinoza et al., 2011), the 2011 drought in Texas (e.g. Long et al., 2013), and the 2003 drought in Europe (e.g. Seitz et al., 2008).

Table 4. Drought events in Europe, Amazon river basin and Texas with corresponding duration taken from literature.

| Region | Year of drought | Considered TWSC months | Examples of literature |
|---|---|---|---|
| Europe | 2003 | June to August | Andersen et al. (2005) |
| | | | Rebetez et al. (2006) |
| | | | Seitz et al. (2008) |
| Amazon river basin | 2005 | May to September | Chen et al. (2009) |
| | | | Frappart et al. (2012) |
| | 2010 | June to September | Espinoza et al. (2011) |
| | | | Frappart et al. (2013) |
| | | | Humphrey et al. (2016) |
| Texas | 2011 | February to October | Humphrey et al. (2016) |
| | | | Long et al. (2013) |

This table contains two new references, which is added to the reference list as follows:

Frappart, F., Papa, F., Santos da Silva, J., Ramillien, G., Prigent, C., Seyler, F. and Calmant, S.: Surface freshwater storage and dynamics in the Amazon basin during the 2005 exceptional drought, Environmental Research Letters, 7(4), 044010, doi:10.1088/1748-9326/7/4/044010, 2012.

Rebetez, M., Mayer, H., Dupont, O., Schindler, D., Gartner, K., Kropp, J. P. and Menzel, A.: Heat and drought 2003 in Europe: a climate synthesis, Annals of Forest Science, 63(6), 569–577, doi:10.1051/forest:2006043, 2006.

**Comment 6.**

[Page10;Ln20] It was not clear to me if the study addressed the last point of this framework's benefit "… identify the most suitable indicator for a specific application".

Response:
We thank the reviewer for this comment. There is a large number of different hydrological regimes for which the TWSC-based indicators would show very different results as soon as e.g., trends are contained in the TWSC time series. Unfortunately, we are not able to elaborate all these applications within this manuscript. With this last point we would like to explain that our aim is to identify strengths and weaknesses of different indicators using our synthetic framework. The identified strengths and weaknesses allow us to decide which indicator might be the most suitable ones (or is not recommended) for a particular application. For example, if the drought period is much shorter than the analyzed TWSC time span and the observations contain a pronounced? trend, we would encourage using indicators like e.g. DSID6 based on the results of our synthetic framework (Sec. 4.1). We modified the corresponding sentence.

Old text:
...; iii) comparing different indicator outputs allows us to identify the most suitable indicator for a specific application.

New text:
...; iii) the synthetic framework enables us to identify strengths and weaknesses of each analysed indicator,  and thereby enables us to choose the most suitable indicator for a specific application.

**Comment 7.**

[Figure 5] I understand that the purpose of this figure is to show the importance of spatial GRACE noise, especially in SA. However, the TWSC time series for EB and WI have low TWSC amplitudes that are equally as low as that of during the simulated drought period in later 2016 (EB) and 2003/2004 (WI). Can you add an explanation to what distinguishes 2005 from these low TWSC?

Response:
Yes, the reviewer is right. The synthetic TWSC for the cluster located in East Brazil (EB) are in later 2016 as low as the TWSC within the simulated drought period in 2005. The same concerns the synthetic TWSC for the cluster located in West India (WI). The synthetic TWSC for WI are in 2003/2004 as low as the TWSC within the simulated drought period. These low TWSC for EB in later 2016 can be explained by the negative acceleration, which was used to generate the synthetic time series. In contrast, the low TWSC for WI in 2003/2004 are based on a positive trend, which has a strong influence here. We do not discuss this influence on the TWSC in detail because we analysed in Sec. 4.1 how trends and accelerations affect drought detection by different indicators.

We hope we addressed this comment by referring the low TWSC in later 2016 in EB and in 2003/2004 in WI to linear trends and constant accelerations.

Old text:
Furthermore, a strong trend and acceleration is contained in the synthesized time series for East Brazil and West India (Tab. 3).

New text:
Furthermore, a strong negative acceleration is contained in the synthesized time series for East Brazil (Tab. 3) leading to strong negative TWSC towards the end of the time series. For West India a strong positive trend leads to  low TWSC at the begin of the time series.

**Comment 8.**

[Section 4.4] I am a bit confused how Figure 11 using real GRACE data is very different from the corresponding figure 10 center, right using the synthetic TWSC. Related to point 5 above, the synthetic data can detect only 2005 drought by design?

Response:
The drought indicators derived by synthetic data should only detect the drought as we designed it, here it was the drought in 2005 but we are able to design different drought duration and magnitudes as we did in one of the experiments described in Sec. 4.3. This drought is by design not equal to the detected real drought that we found in the real GRACE data (Figure 11).

At this point we also need to distinguish the synthetic TWSC data from the real GRACE TWSC data. The synthetic data were computed within a cluster. These clusters are based on regions with similar residual TWSC correlation within the interannual and subseasonal signal. We hypothesized these to be more likely affected by the same hydrological processes, for example by droughts.

We denoted the clusters according to the region where they are located, but the polygons are not exactly equal to, for example, the political boundaries of South Africa, which was used to estimate the results for the real GRACE TWSC. In turn, the polygons of the clusters are not used for the real GRACE TWSC application in Sec. 4.4 because the spatial interpretation of indicators based on political boundaries is better comparable to other research results than to our clusters. Furthermore, we do not intend to compare synthetic data to real data.

To better clarify, that the clusters have specific polygons that are different from the political boundaries, we added an explanation to the methodology part of the framework section.

Old text:
As a result of this procedure, we chose three clusters located in East Brazil (EB), South Africa (SA), and West India (WI), which were also affected by droughts in the past (e.g. Parthasarathy et al., 1987; Rouault and Richard, 2003; Coelho et al., 2016).

New text:
As a result of this procedure, we identified three cluster broadly corresponding to regions in East Brazil (EB, South Africa (SA), and West India (WI) (Fig. 3), which were indeed affected by droughts in the past (e.g. Parthasarathy et al., 1987; Rouault and Richard, 2003; Coelho et al., 2016). A map of all clusters that we identify is provided in Fig. B1; cluster delineations from the above procedure should not be confused with political boundaries or watersheds.

**Comment 9.**

[Page21;Ln20] It is not clear what "simplified noise models" mean. Please elaborate.

Response:
By "simplified noise models" we mean error estimates that do not account for the peculiar way how the GRACE data are collected. For example, by simply assuming globally uniform error does not account for latitude-dependency, density of satellite orbits and data, time dependency of noise levels due to instrument problems or missing data, or the strong error correlation between neighboring grid cells. Here, we exemplarily add one simple example to name one possibility.

Old text:
GRACE studies have been often based on simplified noise models (e.g. Zaitchik et al., 2008; Girotto et al., 2016),; however it is important to account for realistic error and signal correlation, in particular for drought studies where one will push the limits of GRACE spatial resolution.

New text:
GRACE studies have been often based on simplified noise models (e.g. Zaitchik et al., 2008; Girotto et al., 2016), where the GRACE noise model is not derived from the used GRACE data but, for example, from literature and assumed to be spatially uniform and uncorrelated. However, it is important to account for realistic error and signal correlation (Eicker et al. 2014), in particular for drought studies where one will push the limits of GRACE spatial resolution. This signal correlation includes information about, for example, the geographic latitude, the density of the satellite orbits, the time-dependencies of mission periods or North-South-dependencies.

**Comment 10.**

[Page22;Ln10-11] I do not follow "when we did not simulate a trend". When did you?

Response:
Thanks, we modified the corresponding sentence.

Old text:
When we did not simulate a trend, all indicators were able to detect drought, but they identified different timing, duration, and strength.

New text:
When we simulated smaller trends or accelerations, for example for the Southafrican cluster (trend of 4.98 mm/year, accelerations of -0.38 mm^2/year), all indicators were able to detect drought, but they identified different timing, duration, and strength.

**Comment 11.**

[Page23;Ln10] It will be helpful to name the four new indicators (or refer to equations).

Response:
We assume that the reviewer suggested to add the four new indicators to [Page23;Ln20]. Please correct us if we are wrong.

New text:
Four new GRACE-based indicators (DSIA, DSID, DIA and DID) were derived and tested; these are modifications of the above mentioned approaches based on time-accumulated and -differenced GRACE data.

**Minor edits:**

[Equation 19] dot typo?

Response:
The equation is part of the previous sentence. We use the dot in the equation to finish the sentence.

[Table 3] What are the two values for Annual and Semi-annual?

Response:
The coefficients in the table represent the same coefficients as used in Equation 18. In this equation, we have two coefficients for the annual and semi annual signal because these signals are computed using a sine and a cosine wave. So, the values in the table represent b1 and b2 coefficient for annual and c1 and c2 for the semi-annual signal. We included the coefficients in the table and the reference to the equation in the caption of the table.

New caption:
Coefficients (a_0 to c_1 from Eq. 18 and phi_1 from Eq. A1) for signals contained in GRACE-TWSC that were extracted within the clusters of East Brazil, South Africa, and West India. These coefficients are used to simulate synthetic TWSC.

[Page14;Ln16] DSID appears twice.  Thanks, done.
[Page15;Ln1] This sentence is incomplete.

Response:
Please correct us if we are wrong, but we believe the sentence seems incomplete due to the word "results" as a verb instead of a subject and might led to confusion. We replaced it by "derive" to avoid confusion.

Old text:
Applying the Thomas-method to simulated GRACE TWSC results in magnitude, duration and severity of drought, which we show in Fig. 8 for the EB region.

New text:
The Thomas-method is applied to simulated TWSC data to derive magnitude, duration and severity of drought, which we show in Fig. 8 for the EB region.

[Figure6] DSI appears as black line in the plots while legend for DSI is blue. Done, legend is black now.
[Page21;Ln7] GRACE and the DSIA6 → GRACE DSIA6? Thank you.

**Changes in the noise term**

The noise for the synthetic TWSC is derived by using a full variance-covariance matrix. Since this matrix is now derived using the full variance-covariance matrix of the spherical harmonics (computed from normal equations provided by TU GRAZ) instead of using a variance matrix (main diagonal only), the noise levels in Fig. 5, 6, 7, 8, 9 and 10 were slightly updated. However, we would like to emphasize

that these changes do not yield to any changes in our conclusions. Following lines and values have been updated.
O= Old text, N= New text

[Page3;Ln6]
O: ...(2) correlated spatial noise that is related to GRACE, ...
N: ...(2) correlated spatial noise that is related to the peculiar GRACE orbital pattern, ...

[Page15;Ln5]
O: ... up to 28 months (Fig. 8, center) and a severity of about -2500 mm months (Fig. 8, bottom).
N: ... up to 38 months (Fig. 8, center) and a severity of about -4000 mm months (Fig. 8, bottom).

[Page17;Ln2]
O: However, for other cases differences can be more significant, which might lead to misinterpretation (e.g. February and April 2005 for the DI East Brazil, Fig. 9).
N: However, for other cases these differences can be more significant. These may lead to misinterpretation (e.g. May and July 2005 for the DI East Brazil, Fig. 9).

[Page17;Ln14]
O: The DSI shows exceptional drought within the drought period with a maximum of 38 % of the grid cells, i.e. it does not detect exceptional drought in all grid cells.
N: Within the simulated drought period, the DSI indicator identified no more than 14 % of all grid cells as being affected by exceptional drought where it should be 100 %.

[Page20;Ln5]
O: As a reference, the synthetic time series for West India, without any trend or acceleration signal, ranges from about -335 to 76 mm.
N: As a reference, the synthetic time series for West India, without any trend or acceleration signal, ranges from about -323 to 87 mm.

[Page20;Ln14]
O: The severity class with the strongest drought type (i.e. exceptional drought) is only classified by the Zhao- and Houborg-method for East Brazil when using a drought magnitude of -120 mm; this is related to the trend and acceleration signal contained in the simulated TWSC and was already found in Sec. 4.1.
N: Exceptional drought is only classified by the Zhao-method for East Brazil for a simulated drought magnitude of 120 mm; this is related to the trend and acceleration signal contained in the simulated TWSC and was already found in Sec. 4.1.

[Page20;Ln19]
O: Thus, a magnitude of -80 mm in severe drought all applied drought periods (3 to 24 months), while a magnitude of -60 mm leads to moderate dry events and a magnitude of -40 mm to abnormal dry events.
N: Thus, simulating a magnitude of -100 or -120 mm is identified as severe drought for all simulated drought periods (3 to 24 months), while simulating a lower magnitude (-80 mm and -60 mm) moderate or abnormal dry events are identified.

---

## Author Comment (AC2) · 4 Oct 2019

**Author comment to Reviewers comment #2**

Helena Gerdener, Olga Engels and Jürgen Kusche

The author's answers are indicated in red color, as well as old text passages. New text passages are indicated in green color.

**General comments**

In this paper the authors developed a framework that potentially contributes to the understanding of how drought signals propagate through various GRACE drought indicators. By applying three methods (GRACE-based indicators), the authors assessed the skills of newly derived GRACE drought indicators under rather more controlled conditions. This work is significant, as the study is a considerable addition to the existing literature about drought identification methods. Also, the topic is within the scope of Hydrology and Earth System Sciences. Overall, the experimental design is clear, and for the most part, the authors' conclusion are supported by their findings. However, I outline several general concerns, followed by a range of specific comments, which prevent me from recommending this manuscript for publication in its current form. I do hope through that the authors will be able to adequately address my comment and when that is done, this paper should be acceptable for publication.

Response:
Thank you very much for your positive assessment and for your helpful feedback. We hope that we found good solutions to adequately address your comments and to improve the manuscript.

**Comment 1.**

The paper is relatively poorly written. There is a significant number of grammatical/syntactic errors that are present throughout the entire body of the manuscript. I specify several of these in the "Specific Comments" section below, but the authors need to thoroughly check the entire text, as similar or other mistakes may exist elsewhere.

Response:
We thank the reviewer for this comment. The comments in the "Specific Comment" section will be addressed (see below), and we will thoroughly double check the entire text for revision.

**Comment 2.**

Page 3 Line 14 "As can be expected, TWSC and 6 months SPI appear moderately similar (correlation 0.43), characterised by positive peaks at the beginning of 2013. This motivates us to modify common GRACE indicators…" I find the evidence not supportive enough to safely conclude that this link/association between TWSC and SPI is always (or everywhere) the case. The authors should test this on several different regions characterized by varying hydro-climatic conditions. Making such conclusive statements using only one example is scientifically inaccurate.

Response:

We agree with the reviewer that one example is not sufficient to warrant such a conclusive link association between TWSC and the SPI. In fact, we tested this link for other regions, and indeed we found considerable correlations between TWSC and SPI (e.g. Missouri river basin, South Africa, Maharashtra in West India). This was not illustrated (with figures) in the previous version due to space limitations, but we realize we should at least mention these results. Thus, a short sentence about some other regions including correlations is added.

Old text:

As can be expected, TWSC and 6 months SPI appear moderately similar (correlation 0.43), characterised by positive peaks e.g. at the beginning of 2004 and at the end of 2009, and negative peaks at the beginning of 2013. This motivates us to modify common GRACE indicators … .

New text

As can be expected, TWSC and 6 months SPI appear moderately similar (correlation 0.43), characterised by positive peaks e.g. at the beginning of 2004 and at the end of 2009, and negative peaks at the beginning of 2013. We also found correlations between TWSC and 6 months SPI in regions with different hydro-climatic conditions, among others, for the Missouri river basin (0.31), Maharashtra in West India (0.46), and South Africa (0.45). This motivates us to modify common GRACE indicators … .

**Comment 3.**

More information is required for the cluster identification. How exactly were the three clusters determined? The authors also need to clearly specify their exact geographic location.

Response:
We believe that a detailed description of the EM-clustering is given in the literature, so we would like to avoid explaining the EM-algorithm in the main part of the paper. However, we would like to follow the reviewer's suggestion to provide some information to interested readers so we add the main idea and equations of the EM-clustering to the appendix.

Thanks for pointing it out, the information about the polygons can indeed easily be missed out. We adjusted the text and changed the color of the polygons to make them better detectable. We also added the global distribution of all clusters to Fig. B1 in the appendix.

Old text1:
As a result of this procedure, we chose three clusters located in East Brazil (EB), South Africa (SA), and West India (WI), which were also affected by droughts in the past (e.g. Parthasarathy et al., 1987; Rouault and Richard, 2003; Coelho et al., 2016).

New text1:
As a result of this procedure, we identified three clusters located in East Brazil (EB), South Africa (SA), and West India (WI), which were indeed affected by droughts in the past (e.g. Parthasarathy et al.,

1987; Rouault and Richard, 2003; Coelho et al., 2016). Location and shape of the three chosen clusters are shown in Fig. 3 and a global map of all clusters is provided in Fig. B1.

Old text2:
The EM algorithm by Chen (2018) is modified to identify regional clusters by maximizing the likelihood of the data (Alpaydin, 2009).

New text2:
The EM algorithm by Chen (2018) is modified to identify regional clusters. The EM-algorithm alternates expectation and a maximization steps to maximize the likelihood of the data (e.g. Dempster, 1977; Redner, 1984; Alpaydin, 2009). More details about EM-clustering are provided in App. B.

Appendix B: EM-Clustering
Expectation maximization (EM) represents a popular iterative algorithm that is widely used for clustering data. EM partitions data into cluster of different sizes and aims at finding the maximum likelihood of parameters of a predefined probability distribution (Dempster, 1997). In case of a Gaussian distribution the EM-algorithm maximizes the Gaussian mixture parameters, which are the Gaussian mean $\mu_k$, covariance $\Sigma_k$ and mixing coefficients $\pi_k$ (Szeliski 2010). The algorithm then iteratively applies two consecutive steps to maximize the parameters: the expectation step (E-step) and the maximization step (M-step). Within the E-step we estimate the likelihood that a data point $x_t$ is generated from the k-th Gaussian mixture by
E-step:

$$z_{ik} = \frac{1}{Z_i} \pi_k N(x|\mu_k, \Sigma_k) \quad ,$$

The M-step then re-estimates the parameters for each Gaussian mixture:
M-step:

$$\mu_k = \frac{1}{N_k} \sum_i z_{ik} x_i$$

$$\Sigma_k = \frac{1}{N_k} \sum_i z_{ik} (x_i - \mu_k)(x_i - \mu_k)^T$$

$$\pi_k = \frac{N_k}{N}$$

by using the number of points assigned to each cluster via

$$N_k = \sum_i z_{ik} \quad .$$

Using the maximized parameters EM assigns each data point to a cluster. The final global distributed clusters of the AR-parameters (Fig. 3) are shown in Fig. B1. These clusters were derived by modifying and applying an EM-algorithm provided by Chen (2018).

This appendix section contains a new reference, which is added to the reference list as follows:
Szeliski, R.: Computer Vision: Algorithms and Applications, Springer Science and Business Media, 2010

[Figure]

Figure B1. Clusters based on Expectation Maximization (EM) clustering applied to the global autoregressive model (AR)-model coefficients.

**Comment 4.**

The authors should provide more detailed information (characteristics) about specific droughts mentioned in their methodology section.

Response:
To elucidate the chosen drought events, we added a table containing the specific regions and the corresponding considered drought year and TWSC months.

Old text:
Searching for drought duration and magnitude (step 3) led to four droughts seen in GRACE-TWSC: The 2005 and 2010 droughts in the Amazon (e.g. Chen et al., 2009; Espinoza et al., 2011), the 2011 drought in Texas (e.g. Long et al., 2013), and the 2003 drought in Europe (e.g. Seitz et al., 2008).

New text:
Performing literature research for duration and magnitude (step 3) led to four droughts seen in GRACE-TWSC (Tab. 4): The 2005 and 2010 droughts in the Amazon (e.g. Chen et al., 2009; Espinoza et al., 2011), the 2011 drought in Texas (e.g. Long et al., 2013), and the 2003 drought in Europe (e.g. Seitz et al., 2008).

Table 4. Drought events in Europe, Amazon river basin and Texas with corresponding duration taken from literature.

| Region | Year of drought | Considered TWSC months | Examples of literature |
|---|---|---|---|
| Europe | 2003 | June to August | Andersen et al. (2005) |
| | | | Rebetez et al. (2006) |
| | | | Seitz et al. (2008) |
| Amazon river basin | 2005 | May to September | Chen et al. (2009) |
| | | | Frappart et al. (2012) |
| | 2010 | June to September | Espinoza et al. (2011) |
| | | | Frappart et al. (2013) |
| | | | Humphrey et al. (2016) |
| Texas | 2011 | February to October | Humphrey et al. (2016) |
| | | | Long et al. (2013) |

This table contains two new references, which is added to the reference list as follows:

Frappart, F., Papa, F., Santos da Silva, J., Ramillien, G., Prigent, C., Seyler, F. and Calmant, S.: Surface freshwater storage and dynamics in the Amazon basin during the 2005 exceptional drought, Environmental Research Letters, 7(4), 044010, doi:10.1088/1748-9326/7/4/044010, 2012.

Rebetez, M., Mayer, H., Dupont, O., Schindler, D., Gartner, K., Kropp, J. P. and Menzel, A.: Heat and drought 2003 in Europe: a climate synthesis, Annals of Forest Science, 63(6), 569–577, doi:10.1051/forest:2006043, 2006.

**Specific Comments**

Abstract
"Thus, this study aims at a better understanding of how drought signals, in the presence of trends and GRACE-specific spatial noise, propagate through GRACE drought indicators": This phrase is perhaps the essence of the abstract; therefore it should be able to provide the necessary information on its own. The authors need to specify which trends they are referring to.

Response:
Thanks, we are referring to linear trends and constant accelerations in the paper, which are described with a linear term $a_1 (t - t_0)$ and a quadratic term $a_2 \frac{1}{2} (t - t_0)^2$ in Eq. 18. Linear trends and possible constant accelerations in GRACE TWSC can result from many different hydrological processes, for example, accelerations can results from linear trends in the fluxes precipitation, evapotranspiration and runoff. To specify the terms, we added linear trend and constant accelerations to the abstract.

New text:
Thus, this study aims at a better understanding of how drought signals propagate through GRACE drought indicators in the presence of linear trends, constant accelerations and GRACE-specific spatial noise.

According to this comment, we specified the meaning of trends and accelerations for the subsequent usage of the terms.

Page 7 Line 16
O: The signal is computed by ... at time t with a constant $a_0$, linear trend $a_1$ and acceleration $a_2$ terms, an annual signal $b_1$ and $b_2$, and similar for a semi-annual signal c_1 and c_2.

N: The signal is computed by ... at time t with a constant $a_0$, linear trend $a_1$ and constant acceleration $a_2$ terms, an annual signal $b_1$ and $b_2$, and similar for a semi-annual signal c_1 and c_2. Trends and possible accelerations in GRACE TWSC can result from many different hydrological processes. For example, accelerations can results from trends in the fluxes precipitation, evapotranspiration and runoff (e.g. Eicker et al. 2016). In the following, the linear trends are denoted as trends and constant accelerations are denoted as accelerations.

Line 10 application-dependent Yes, corrected, thanks.
Line 10 large differences Corrected.
Line 11 particularly Addressed.
Line 12 We show that trend and accelerations – what do the authors mean by "accelerations"?

Response:
We mean possible constant accelerations contained in the analysed time series that is described by the quadratic term $a_2 \frac{1}{2} (t - t_0)^2$ in Eq. 18. We hope this is more clear now by specifying the trends, as the reviewer recommended in the first comment of the "Specific Comments" section (above).

Line 17 affect the Done, thanks.
Line 18 replace "reach" with "range" Done.
Line 24 led Yes, thanks, corrected.

Line 4 depends on the accumulation period considered – unclear

Response:
Yes, we see that the term accumulation period leads to confusion here, because it is introduced at a later point. We remove this part of the sentence.

Old text:
For South Africa, due to a complex rainfall regime, areas and percentage of land surface affected by drought can vary strongly (Rouault and Richard, 2005) and their identification depends on the accumulation period considered.

New text:
For South Africa, due to a complex rainfall regime, areas and percentage of land surface affected by drought can vary strongly (Rouault and Richard, 2005).

Line 16 Much fewer Done.

Line 23 and the first data are expected

Response:
We updated this sentence, because the first data is now available and not "expected to become available in May 2019".

Old text:
Meanwhile, GRACE has been continued with the GRACE-FO mission and the first data are expected to become available in May 2019.

New text:
Meanwhile, GRACE has been continued with the GRACE-FO mission and the first data are available.

Line 27 they found good agreement to net precipitation minus evaporation. - unclear

Response:
We agree this needs clarification. The agreement between TWSC and the combination of the net precipitation and evaporation is meant.

Old text:
For example, Seitz et al. (2008) investigated the 2003 heat wave over seven Central European basins using GRACE timeseries; they found good agreement to net precipitation minus evaporation.

New text:
For example, Seitz et al. (2008) investigated the 2003 heat wave over seven Central European basins using GRACE timeseries; they found a good agreement between TWSC and the combination of net precipitation and evaporation.

Line 34 without utilizing external information – please specify

Response:
Separating a specific compartment from GRACE TWSC data requires knowledge from other observation techniques or model outputs, because GRACE can only measure the sum of all compartments.

Old text:
However, neither GRACE nor GRACE-FO enable one to separate different compartments such as groundwater storage without utilizing external information, and their spatial (about 300 km for GRACE) and temporal (nominally one month) resolution are limited.

New text:
However, neither GRACE nor GRACE-FO enable one to separate different storage compartments such as groundwater storage without utilizing additional (e.g. compartment-specific) observations or model outputs, and their spatial (about 300 km for GRACE) and temporal (nominally one month) resolution are limited.

Line 4 delete "e.g." Done, thanks.
Line 7 "smoothing" Done.
Line 17 What are "differencing periods"

Response:
We agree that the term here is confusing, because it was not introduced before. We change the sentence.

Old text:
This motivates us to modify common GRACE indicators to account for accumulation and differencing periods.

New text:
This motivates us to modify common GRACE indicators to account for accumulation periods of input data, which are used with e.g. 6 months SPI, but also periods that are based on differences of input data.

Line 21 spatially averaged Done, thanks.
Line 26 will complete the paper Done.

Line 2 explore Thanks, corrected.
Line 10 more regularly Corrected.

Line 10 we construct Done, thanks.
Line 13 including the introduced (in Sec. 2.3) signal … Done.
Line 26 … following A et al. (2013) … is there something missing here?

Response:
Indeed it might lead to confusion but A is the full last name.

Line 8 drought onset and end Corrected.
Lines 10-14 these thresholds are rather arbitrarily made. It seems to me that a single value for the drought duration and magnitude should not be used for different hydrologic regimes.

Response:
We do not agree with the reviewer that these values for the threshold are arbitrary because we identified these values by analysing different historical droughts that were detected in literature using GRACE TWSC. Of course, one can not assume that one value for drought duration and magnitude can be detected in different hydrological regimes, but this is not what we intended with this analysis. We aim at simulating a signal that is similar to existing drought signals contained in GRACE, which is able to show up as exceptional drought in at least one indicator.

Line 5 inappropriate use of English for a scientific paper Corrected.

Old text
However, seen these difficulties, we decided to stick to the most simple TWSC drought model, i.e. a constant water storage deficit within a given time span.

New text
However, due to these difficulties, we decided to use the most simple TWSC drought model, i.e. a constant water storage deficit within a given time span.

Line 10 delete "would" Corrected, thanks.

Line 17 for the 3, and 6 months differenced DSID Sorry we do not see a difference.

Line 24 climatic phenomenon Yes, thanks, corrected.
Line 24 delete "related to climatic conditions" as it is redundant Corrected.

Line 9 in the northeastern Thanks, we changed it to "Northeastern".

Due to this comment we also changed following text:

Old text:
Fig. 3 shows the estimated AR-model coefficients, which represent the temporal correlations, ranging from very low up to 0.3, e.g. over the Sahara or in South West Australia, to about 0.8, for example in Brazil or in South Eastern U.S. EM-clustering is then based on these coefficients.

New text:
Fig. 3 shows the estimated AR-model coefficients, which represent the temporal correlations, ranging from very low up to 0.3, e.g. over the Sahara or in South West Australia, to about 0.8, for example in Brazil or in Southeastern U.S. EM-clustering is then based on these coefficients.

Line 22 particularly Done.
Line 25 the onset and end Done.

---

## Author Comment (AC3) · 4 Oct 2019

We thank the editor Bettina Schaefli for the opportunity to answer the reviewers comments and we also thank the reviewers for the very helpful comments they provided. These comments certainly helped us to improve the manuscript and we are very grateful that our paper attracted interest. We addressed the reviewers comments in our author comments. The reviewer's comments were answered in the proposed way: Referee comment, author's answer, and changes in the manuscript. Red color indicates our answers towards the referee comments and, if necessary, the old sentence/text to what the comment refers. Green color indicates the new changes in the text. We hope that we could adequately address all considered aspects.